# MULTI-SCALE FEATURE LEARNING DYNAMICS: INSIGHTS FOR DOUBLE DESCENT

## ABSTRACT

A key challenge in building theoretical foundations for deep learning is the complex optimization dynamics of neural networks, resulting from the high-dimensional interactions between the large number of network parameters. Such non-trivial dynamics lead to intriguing model behaviors such as the phenomenon of "double descent" of the generalization error. The more commonly studied aspect of this phenomenon corresponds to *model-wise* double descent where the test error exhibits a second descent with increasing model complexity, beyond the classical U-shaped error curve. In this work, we investigate the origins of the less studied *epoch-wise* double descent in which the test error undergoes two non-monotonous transitions, or descents as the training time increases. By leveraging tools from statistical physics, we study a linear teacher-student setup exhibiting epoch-wise double descent similar to that in deep neural networks. In this setting, we derive closed-form analytical expressions for the evolution of generalization error over training. We find that double descent can be attributed to distinct features being learned at different scales: as fast-learning features overfit, slower-learning features start to fit, resulting in a second descent in test error. We validate our findings through numerical experiments where our theory accurately predicts empirical findings and remains consistent with observations in deep neural networks.

## 1 INTRODUCTION

Classical wisdom in statistical learning theory predicts a trade-off between the generalization ability of a machine learning model and its complexity, with highly complex models less likely to generalize well (Friedman et al., 2001). If the number of parameters measures complexity, deep learning models sometimes go against this prediction (Zhang et al., 2016): deep neural networks trained by stochastic gradient descent exhibit a so-called *double descent* behavior (Belkin et al., 2019b) with increasing model parameters. Specifically, with increasing complexity, the generalization error first obeys the classical U-shaped curve consistent with statistical learning theory. However, a second regime emerges as the number of parameters is further increased past a transition threshold where generalization error drops again, hence the "double descent" or more accurately *model-wise double descent* (Nakkiran et al., 2019).

Nakkiran et al. (2019) showed that the phenomenon of double descent is not limited to varying model size but is also observed as a function of training time or epochs. In this case as well, the so-called *epoch-wise double descent* is in apparent contradiction with the classical understanding of over-fitting (Vapnik, 1998), where one expects that longer training of a sufficiently large model beyond a certain threshold should result in over-fitting. This has important implications for practitioners and raises questions about one of the most widely used regularization method in deep learning (Goodfellow et al., 2016): early stopping. Indeed, while one might expect early stopping to prevent over-fitting, it might in fact prevent models from being trained at their fullest potential.

Since the 1990s, there has been much interest in understanding the origins of non-trivial generalization behaviors of neural networks (Opper, 1995; Opper & Kinzel, 1996). The authors of Krogh & Hertz (1992b) were among the first to provide theoretical explanations for (model-wise) double descent in linear models. Summarily, at intermediate levels of complexity, where the model size is equal to the number of training examples, the model is very sensitive to noise in training data and hence, generalizes poorly. This sensitivity to noise reduces if the model complexity is either de-

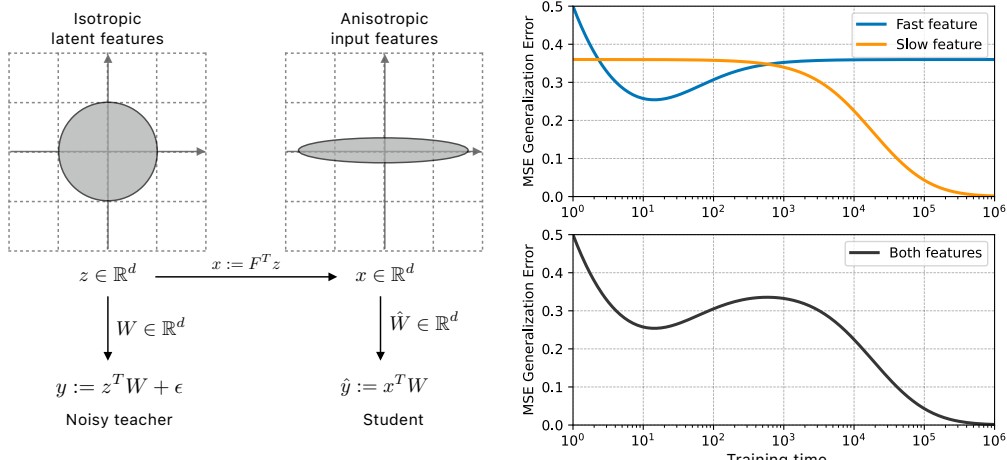

Figure 1: **Left**: The teacher is the data generating process that operates on isotropic Gaussian inputs $z$. The student is trained on a dataset generated by the teacher, $\mathcal{D} = \{x_i, y_i\}_{i=1}^{n}$ where $x := F^T z$ follow an anisotropic Gaussian distribution such that the directions with larger/smaller variance are learned faster/slower. The condition number of $F$ determines how much faster some features are learned than the others. One can think of $z$ as the latent factors of variation on which the teacher operates, while $x$ can be thought as the pixels that the student learns from. **Right**: The generalization error as the training time proceeds. (top): The case where only the fast-learning feature or slow-learning feature are trained. (bottom): The case where both features. Features that are learned on a faster time-scale are responsible for the classical U-shaped generalization curve, while the second descent can be attributed to the features that are learned at a slower time-scale.

creased or increased. More recently, the double descent phenomena has been also studied for more complex models such as two-layer neural networks and random feature models (Ba et al., 2019; Mei & Montanari, 2019; D'Ascoli et al., 2020; Gerace et al., 2020).

The majority of previous work in this direction focuses on understanding the *asymptotic* behavior of model performance, i.e., where training time $t \to \infty$. In recent years, there has been an interest in studying the *non-asymptotic* (finite training time) performance, suggesting that several intriguing properties of neural networks can be attributed to different features being learned at different scales. Among the limited work studying the particular epoch-wise double descent, Nakkiran et al. (2019) introduces the notion of *effective model complexity* and hypothesizes that it increases with training time and hence unifies both model-wise and epoch-wise double descent. Through a combination of theory and empirical results, Heckel & Yilmaz (2020) find that the dynamics of evolution of single and two layer networks under gradient descent, can be perceived to be the superposition of two bias/variance curves with different minima times, thus leading to non-monotonic test error curves.

In this work, we build on Bös et al. (1993); Bös (1998); Advani & Saxe (2017); Mei & Montanari (2019) which analyze *model-wise* double descent through the lens of linear models, to probe the origins of *epoch-wise* double descent. Particularly,

- We introduce a linear teacher-student model which, despite its simplicity, exhibits some of intriguing properties of generalization dynamics in deep neural networks. (Section 2.1)
- In the limit of high dimensions, we leverage the replica method developed in statistical physics to derive closed-form expressions for the generalization dynamics of our teacher-student setup, as a function of training time and regularization strength. (Section 2.2)
- Consistent with recent findings, we provide an explanation for the existence of epoch-wise double descent through the lens of multi-scale feature learning. (Figure 1)
- We perform simulation experiments to validate our analytical predictions. We also conduct experiments with deep networks, showing that our teacher-student setup exhibits generalization behavior which is qualitatively similar to that of deep networks. (Figure 2)

## 2 ANALYTICAL RESULTS

Stochastic Gradient Descent (SGD) — the de facto optimization algorithm for neural networks — exhibits complex dynamics arising from a large number of parameters (Kunin et al., 2020). However, it is possible to describe some aspects of the high-dimensional *microscopic* dynamics of neural networks in terms of low-dimensional understandable *macroscopic* entities. In a series of seminal papers by Gardner (Gardner, 1988; Gardner & Derrida, 1988; 1989), the *replica method* of statistical physics was adopted to derive expressions describing the generalization behavior of large linear models trained using SGD. In this paper, we employ Gardner's analysis to build upon an established line of work studying linear and generalized linear models (Seung et al., 1992; Kabashima et al., 2009; Krzakala et al., 2012). While most of previous work study the asymptotic ($t \to \infty$) generalization behavior, we adapt these methods to study transient learning dynamics of generalization for finite training time. In the following, we first introduce a teacher-student model that exhibits interesting characteristics of modern neural networks. We then adapt the replica method to study the generalization performance as a function of training time and amount of regularization.

### 2.1 A TEACHER-STUDENT SETUP

**Teacher:** We study a supervised linear regression problem in which the training labels $y$, are generated by a noisy linear model (Figure 1),

$$y := y^* + \epsilon, \qquad y^* := \boldsymbol{z}^T W, \qquad z_i \sim \mathcal{N}(0, \frac{1}{\sqrt{d}}), \tag{1}$$

where $\boldsymbol{z} \in \mathbb{R}^d$ is the teacher's input and $y^*, y \in \mathbb{R}$ are the teacher's noiseless and noisy outputs, respectively. $W \in \mathbb{R}^d$ represents the (fixed) weights of the teacher and $\epsilon \in \mathbb{R}$ is the noise. Both $W$ and $\epsilon$ are drawn i.i.d. from Gaussian distributions with zero means and variances of 1 and $\sigma_\epsilon^2$, respectively.

**Student:** A student model is correspondingly chosen to be a similar shallow network with trainable weights $\hat{W} \in \mathbb{R}^d$. The student model is trained on $n$ training pairs $\{(\boldsymbol{x}^\mu, y^\mu)\}_{\mu=1}^n$, with the labels $y^\mu$ being generated by the above teacher network, as,

$$\hat{y} := \boldsymbol{x}^T \hat{W}, \qquad s.t. \qquad \boldsymbol{x} := F^T \boldsymbol{z}, \tag{2}$$

where the matrix $F \in \mathbb{R}^{d \times d}$ is a predefined and fixed **modulation matrix** regulating the student's access to the true input $\boldsymbol{z}$. One can think of $\boldsymbol{z}$ as the latent factors of variation on which the teacher operates, while $\boldsymbol{x}$ can be thought as the pixels that the student learns from.

**Learning paradigm:** To train our student network, we use stochastic gradient descent (SGD) on the regularized mean squared loss, evaluated on the $n$ training examples as,

$$\mathcal{L}_{\mathcal{T}} := \frac{1}{2n} \sum_{\mu=1}^n (y^\mu - \hat{y}^\mu)^2 + \frac{\lambda}{2} ||\hat{W}||_2^2 \tag{3}$$

where $\lambda \in [0, \infty)$ is the regularization coefficient. Optimizing Eq. 3 with stochastic gradient descent (SGD) yields the typical update rule,

$$\hat{W}_t \leftarrow \hat{W}_{t-1} - \eta \nabla_{\hat{W}} \mathcal{L}_{\mathcal{T}} + \xi, \tag{4}$$

in which $t$ denotes the training step and $\eta$ is the learning rate. Additionally, $\xi \sim \mathcal{N}(0, \sigma_\xi^2)$ models the stochasticity noise of the optimization algorithm (Bottou et al., 1991).

**Macroscopic variables:** The quantity of interest in this work, is the expected generalization error of the student, determined by averaging the student's error over all possible input-target pairs and noise realizations, as,

$$\mathcal{L}_{\mathcal{G}} := \frac{1}{2} \mathbb{E}_z \big[ (y^* - \hat{y})^2 \big]. \tag{5}$$

As shown in Bös et al. (1993), if $n, d \to \infty$ with a constant ratio $\frac{n}{d} < \infty$, Eq. 5 can be written as a function of two macroscopic scalar variables $R, Q \in \mathbb{R}$,

$$\mathcal{L}_{\mathcal{G}} = \frac{1}{2} (1 + Q - 2R), \tag{6}$$

where $\sigma_\epsilon^2$ is the variance of the teacher's output noise and,

$$R := \frac{1}{d} W^T F \hat{W}, \qquad Q := \frac{1}{d} \hat{W}^T F^T F \hat{W}, \tag{7}$$

See App. B.1 for the proof.

Both $R$ and $Q$ have clear interpretations; $R$ is the dot-product between the teacher's weights $W$ and the student's *modulated* weights $F\hat{W}$, hence can be interpreted as the **alignment between the teacher and the student**. Similarly, $Q$ can be interpreted as the **student's modulated norm**. The negative sign of $R$ in Eq. 6 suggests that the larger $R$ is, the smaller the generalization error gets. At the same time, $Q$ appears with a positive sign suggesting the students with smaller (modulated) norm generalize better.

As a remark, note that both $R$ and $Q$ are functions of $\hat{W}$ which itself is a function of training iteration $t$ and the regularization strength $\lambda$. Therefore, hereafter, we denote the above quantities as $\mathcal{L}_\mathcal{G}(t, \lambda)$, $R(t, \lambda)$, and $Q(t, \lambda)$.

## 2.2 MAIN RESULTS

In this Section, we present our main analytical results, with Section 2.3 containing a sketch of our derivations. For brevity of the results, here, we only present the results for $\sigma_\epsilon^2 = \lambda = 0$. See App. B for the general case and the detailed proofs.

**General matrix $F$.** Let $Z := [z^\mu]_{\mu=1}^n \in \mathbb{R}^{n \times d}$ and $X := [x^\mu]_{\mu=1}^n \in \mathbb{R}^{n \times d}$ denote the input matrices for the teacher and student such that $X := ZF$. For a general modulation matrix $F$, the input covariance matrix has the following singular value decomposition (SVD),

$$X^T X = F^T Z^T Z F = V \Lambda V^T, \tag{8}$$

in which the diagonal matrix $\Lambda$ contains the eigenvalues of the student's input covariance matrix. Solving the dynamics of gradient descent as in Eq. 4, we arrive at the following exact analytical expressions for $R(t)$ and $Q(t)$,

$$R(t) = \frac{1}{d} \mathbf{Tr}(D) \qquad \text{where,} \quad D := \left(I - [I - \eta \Lambda]^t\right), \tag{9}$$

$$Q(t) = \frac{1}{d} \mathbf{Tr}\left(A^T A\right) \quad \text{where,} \quad A := FVDV^T F^{-1}, \tag{10}$$

in which $\mathbf{Tr}(.)$ is the trace operator. See App. B.2 the proof.

*Remark:* The solution in Eqs. 9 and 10 are exact, however, they require the empirical computation of the eigenvalues $\Lambda$. Below, we treat a special case of the dynamics that allow us to derive approximate solutions that do not explicitly depend on $\Lambda$.

**Special case: Fast and slow features.** We now study a case where the modulation matrix $\boldsymbol{F}$ has a specific structure described in Assumption 1.

**Assumption 1.** *The modulation matrix, $F$, under a SVD, $F := U\Sigma V^T$ has two sets of singular values such that the first $p$ singular values are equal to $\sigma_1$ and the remaining $d - p$ singular values are equal to $\sigma_2$. We let the condition number of $F$ to be denoted by $\kappa := \frac{\sigma_1}{\sigma_2} > 1$.*

By employing the replica method of statistical physics (Gardner, 1988; Gardner & Derrida, 1988), we now derive approximate expressions for $R(t)$ and $Q(t)$. To begin with, we first define the following auxiliary variables,

$$\alpha_1 := \frac{n}{p}, \; \alpha_2 := \frac{n}{d-p}, \qquad \tilde{\lambda}_1 := \frac{d}{p} \frac{1}{\eta \sigma_1^2 t}, \; \tilde{\lambda}_2 := \frac{d}{d-p} \frac{1}{\eta \sigma_2^2 t}, \tag{11}$$

and also let,

$$a_i = 1 + \frac{2\tilde{\lambda}_i}{(1 - \alpha_i - \tilde{\lambda}_i) + \sqrt{(1 - \alpha_i - \tilde{\lambda}_i)^2 + 4\tilde{\lambda}_i}}, \qquad \text{for} \qquad i \in \{1, 2\}. \tag{12}$$

The closed-from scalar expression for $R(t)$ is then given by,

$$R(t) = R_1 + R_2, \quad \text{where,} \quad R_1 := \frac{n}{a_1 d}, \quad \text{and,} \quad R_2 := \frac{n}{a_2 d} \tag{13}$$

For $Q(t)$, we accordingly define two more auxiliary variables,

$$b_i = \frac{\alpha_i}{a_i^2 - \alpha_i}, \quad c_i = 1 - 2R_i - \frac{n}{d}\frac{2 - a_i}{a_i} \quad \text{for} \quad i \in \{1, 2\}, \tag{14}$$

with which the closed-from scalar expression for $Q(t)$ reads,

$$Q(t) = Q_1 + Q_2, \quad \text{where,} \quad Q_1 := \frac{b_1 b_2 c_2 + b_1 c_1}{1 - b_1 b_2}, \quad \text{and,} \quad Q_2 := \frac{b_1 b_2 c_1 + b_2 c_2}{1 - b_1 b_2}. \tag{15}$$

By plugging Eqs. 13 and 15 into Eq. 6, one obtains a closed-form expression for $\mathcal{L}_\mathcal{G}(t)$ as a function of the training time. See App. B.3 for the proof.

*Remark:* Eq. 11 indicates that the singular values of $F$, are directly multiplied by $t$. That implies that the learning speed of each feature is scaled by the magnitude of its corresponding singular value. As an illustration, the figure on the right shows the evolution of $R_1$, $R_2$, and $R = R_1 + R2$ for a case where $p = d/2$, $\sigma_1 = 1$, and $\sigma_1 = 0.01$ implying a condition number of $\kappa = 100$.

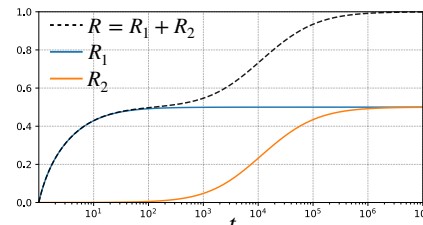

### 2.3 SKETCH OF DERIVATIONS

In this Section, we sketch the key steps in the derivation of our main results. For the sake of simplicity, here we only treat the case where $\sigma_\epsilon = \lambda = 0$. The general case with detailed proofs are presented in App B.

**Exact dynamics of SGD.** Recall the gradient descent update rule in Eq. 4. For the linear model defined in Eqs. 1-2, learning is governed by the following discrete-time dynamics,

$$\hat{W}_t = \hat{W}_{t-1} - \eta \nabla_{\hat{W}_{t-1}} \mathcal{L}_\mathcal{T}, \tag{16}$$

$$= \hat{W}_{t-1} - \eta \big[ - X^T (y - X\hat{W}_{t-1}) \big]. \tag{17}$$

With the assumption that $\hat{W}_{t=0} = \mathbf{0}$, the dynamics admit the following exact closed-form solution,

$$\hat{W}_t = \Big( I - \big[ I - \eta X^T X \big]^t \Big) (X^T X)^{-1} X^T y := \tilde{W}(t). \tag{18}$$

With a SVD on $X^T X$, Eqs. 9-10 can then be obtained by substituting $\hat{W}_t$ in Eqs. 7. As a remark, note that one can recover the results of Advani & Saxe (2017) by setting $F = I$. In that case, the eigenvalues of $X^T X$ follow a Marchenko–Pastur distribution (Marchenko & Pastur, 1967).

**Induced probability density of SGD.** It is well-known (Kuhn & Bos, 1993; Solla, 1995) that probability distribution of weight configurations for network weights $\hat{W}$ trained via SGD on a loss $\mathcal{L}(\hat{W})$, tend to the Gibbs distribution such that,

$$P(\hat{W}) = \frac{1}{Z_\beta} e^{-\beta \mathcal{L}(\hat{W})}, \tag{19}$$

in which $Z_\beta$ is the partition function $\Big( \int d\hat{W} \exp(-\beta \mathcal{L}(\hat{W})) \Big)$ and $\beta$ is called the *inverse temperature* and is inversely proportional the stochastic noise of SGD, $\xi$, defined in Eq. 4. Intuitively, for small $\beta$, the distribution of $P(\hat{W})$ is almost uniform, while as $\beta \to \infty$, $P(\hat{W})$ becomes more concentrated around the minimum of the training loss.

It is important to highlight that Eq. 19 describes the *equilibrium* distribution of the student network's weights, i.e., at the end of training ($t \to \infty$). However, we are interested in studying the trajectory

of student's weights *during* the course of training, i.e., for finite $t$. To that end, we derive the **time-dependent** probability density over $\hat{W}$,

$$P(\hat{W}, t) = \frac{1}{Z_{\beta}, t} e^{-\beta \tilde{\mathcal{L}}(\hat{W}, t)}, \quad \text{where,} \tag{20}$$

$$\tilde{\mathcal{L}}_T(\hat{W}, t) := \frac{1}{2n} \sum (\hat{y}^\mu - \tilde{y}^\mu(t))^2 + \frac{\lambda}{2} ||\hat{W}||_2^2, \tag{21}$$

$$= \frac{1}{2n} \sum (\hat{y}^\mu - x^{\mu^T} \tilde{W}(t))^2 + \frac{\lambda}{2} ||\hat{W}||_2^2, \quad (\tilde{W}(t) \text{ defined in Eq. 18}) \tag{22}$$

$$\approx \mathcal{L}_T(\hat{W}) + \frac{1}{2}\big(\lambda + \frac{1}{\eta t}\big) ||\hat{W}||_2^2. \tag{23}$$

*Remark:* $\tilde{\mathcal{L}}_T(\hat{W}, t)$ is a modified loss such that its minimum (equilibrium distribution) is achieved at the $t^{\text{th}}$ iterate of gradient descent on $\mathcal{L}(\hat{W})$. The schematic diagram on the right illustrates this equivalence, such that, $\arg\min_{\hat{W}} \tilde{\mathcal{L}}_T(\hat{W}, t) = \hat{W}_t$, where $\hat{W}_t$ is the defined in Eq. 4.

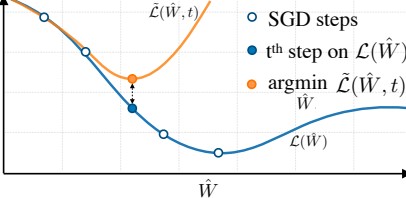

**The typical generalization error.** To determine the *typical* generalization performance at time $t$, one proceeds by first computing the free-energy of the system as,

$$f := -\frac{1}{\beta d} \mathbb{E}_{W,z}\big[\ln Z_{\beta, t}\big]. \tag{24}$$

Free-energy is a self-averaging property where its *typical/most probable* value coincides with its *average* over proper probability distributions Engel & Van den Broeck (2001). Therefore, to determine the typical values of $R$ and $Q$, we extremize the free-energy w.r.t. those variables.

Due to the logarithm inside the expectation, analytical computation of Eq. 24 is intractable. However, the replica method (Mézard et al., 1987) allows us to tackle this through the following identity,

$$\mathbb{E}_{W,z}[\ln Z_{\beta, t}] = \lim_{r \to 0} \frac{\mathbb{E}_{W,z}[Z_{\beta, t}^r] - 1}{r}. \tag{25}$$

Computation of the free-energy via replica method and its subsequent extremization w.r.t $R$ and $Q$, we arrive at Eqs. 13 and 15. See App. B.3 for more details.

To summarize, using the replica method, we are able to cast the high-dimensional dynamics of SGD into simple scalar equations governing $R$ and $Q$ and, consequently, the generalization error $\mathcal{L}_G$. While our analysis is limited to the specific teacher and student setup, this simple model already exhibits dynamics qualitatively similar to those observed in more complex networks, as we now illustrate.

## 3 EXPERIMENTAL RESULTS

In this Section, we conduct numerical simulations to validate our analytical results and provide clear insights on the macroscopic dynamics of generalization. We also conduct experiments on real-world neural networks showing a close qualitative match between the generalization behavior of neural networks and our teacher-student setup.

**For real-world experiments**, we train a **ResNet18** (He et al., 2016) with large layer widths $[64, 2 \times 64, 4 \times 64, 8 \times 64]$. We follow the training setup of Nakkiran et al. (2019); Label noise with a probability $0.15$ randomly assign an incorrect label to training examples. Noise is sampled only once before the training starts. We train using Adam (Kingma & Ba, 2014) with learning rate of $1e - 4$ for 1K epochs. Real-world experiments are averaged over 50 random seeds. To ensure reproducibility, we include the complete source code in a `GitHub repository` as well as an anonymous `Collab notebook`.

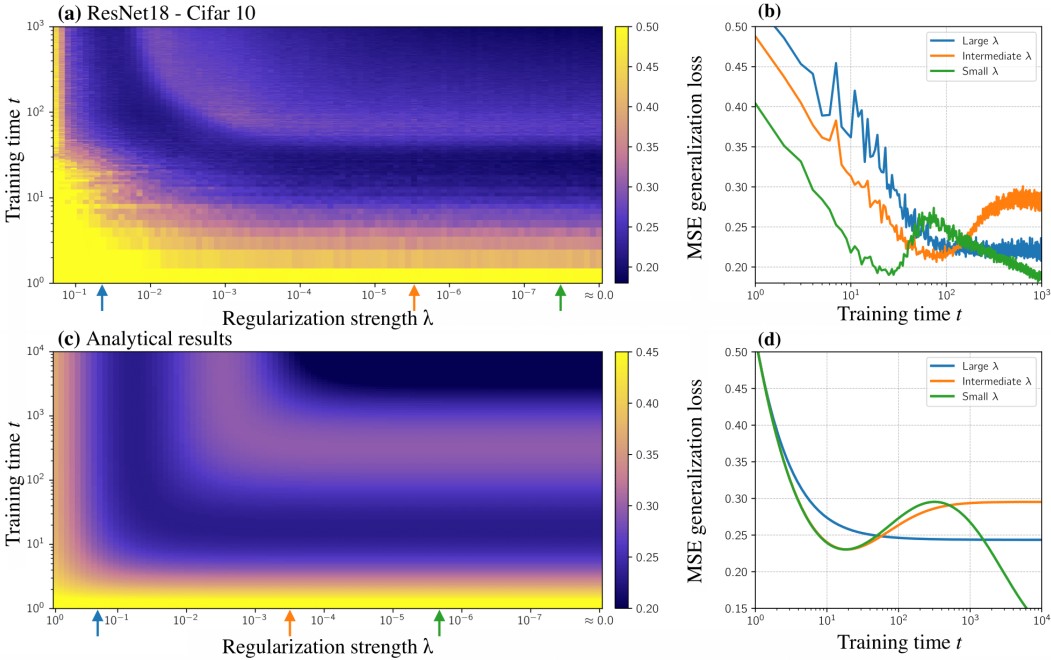

Figure 2: **A qualitative comparison between a ResNet-18 and our analytical results. (a)**: Heat-map of empirical generalization error (0-1 classification error) for the ResNet-18 trained on CIFAR-10 with 15% label noise. X-axis denotes the inverse of weight-decay regularization strength and Y-axis represents the training time. **(c)**: Heat-map of the analytical generalization error (mean squared error) for the linear teacher-student setup with $\kappa = 100$, the condition number of the modulation matrix. **(b, d)**: Three slices of the heat-maps for large, intermediate, and small amounts of regularization. **Analysis**: As predicted by Eqs. 13 and 15, $\kappa = 100$ implies that a subset of features are learned 100 times faster that the rest. Intuitively, large amounts of regularization allow for the fast-learning features to be learned by not to overfit. Intermediate levels of regularization result in a classical U-shaped generalization curve but prevent slow features from learning. Small amounts of regularization allow for both fast and slow features to be learned, leading to double descent.

## 3.1 MATCH BETWEEN THEORY AND REAL-WORLD EXPERIMENTS

We conduct an experiment on the classification task of CIFAR-10 (Krizhevsky et al., 2009) with varying amount of weight decay regularization strength $\lambda$. We monitor the generalization error (0-1 test error) during the course of training and visualize a heat-map of the generalization error for different $\lambda$'s in Figure 2 (a).

We also conduct a similar experiment with the teacher-student setup presented in Section 2.1. We visualize a heat-map of the generalization error which is the mean squared error (MSE) over test distribution in Figure 2 (b). Particularly, we plot Eqs. 13 and 15 with a constant $\kappa = 100$. As a remark, we note that a $\kappa = 100$ implies that a subset of features are learned 100 times faster than other features.

It is observed that in both experiments, a model with intermediate levels of regularization displays a typical overfitting behavior where the generalization error decreases first and then overfits. This is consistent with Eq. 87 which indicates larger amounts of regularization prevent slow feature from being learned as $\lambda$ and the inverse of $t$ are summed. In other words, learning of slow features requires large weights, something that is penalized by the weight-decay. On the other hand, a model with smaller amount of regularization exhibits the double descent generalization curve.

We also validate our derived analytical expressions by running numerical simulations which are presented in Figure 4.

## 3.2 THE PHASE DIAGRAM

To further investigate the transition between the two phases of *classical single descent* and *double descent*, we explore the phase diagram. Recall that with Eq. 6, one can fully characterize the evolution of the generalization dynamics in terms of two scalar variables instead of the $d$-dimensional parameter space. $R$ and $Q$ presented in Eq. 7 are macroscopic variables where $R$ represents **the alignment between the teacher and the student** and $Q$ is the **student's (modulated) norm**. Hence, a better generalization performance is achieved with larger $R$ and smaller $Q$.

$R$ and $Q$ are not free parameters and both depend on the training dynamics through Eqs. 13 and 15. Nevertheless, it is instructive to visualize the generalization error for all pairs of $(R, Q)$. In Figure 3, we visualize the $RQ$-plane for $(R, Q) \in [0.0, 0.8] \times Q \in [0.0, 1.6]$. At the time of initialization, $(R, Q) = (0, 0)$ as the models are initialized at the origin. As training time proceeds, values of $R$ and $Q$ follow the depicted trajectories. In Figure 3, different trajectories correspond to different values of $\kappa$, the condition number of the modulation matrix $\boldsymbol{F}$ in Eq. 2. It is important to note that *the closer a trajectory is to the lower-right, the better the generalization error gets.*

The yellow curve which corresponds to the case with large $\kappa = 1e5$ meaning that a subset of features are extremely slower than the others that practically do not get learned. In that case, generalization error exhibits traditional over-fitting due to over-training. On the phase diagram, the yellow trajectory starts at $(0, 0)$ and moves towards Point $A$ which has the lowest generalization error of this curve. Then as the training continues, $Q$ increases and as $t \to \infty$ the trajectory lands at Point $B$ which has the worse generalization error. The curves in orange, green and blue correspond to trajectories with $\kappa = 1e3$, $\kappa = 1e2$, $\kappa = 1e1$, respectively. They follow the case of $\kappa = 1e5$ up to the vicinity of Point B, but then the trajectories slowly incline towards another fixed point, Point $C$ signalling a second descent in the generalization error.

The phase diagram along with the corresponding generalization curves in Figure 2 illustrate that features that are learned on a faster time-scale are responsible for the initial conventional U-shaped generalization curve, while the second descent can be attributed to the features that are learned at a slower time-scale.

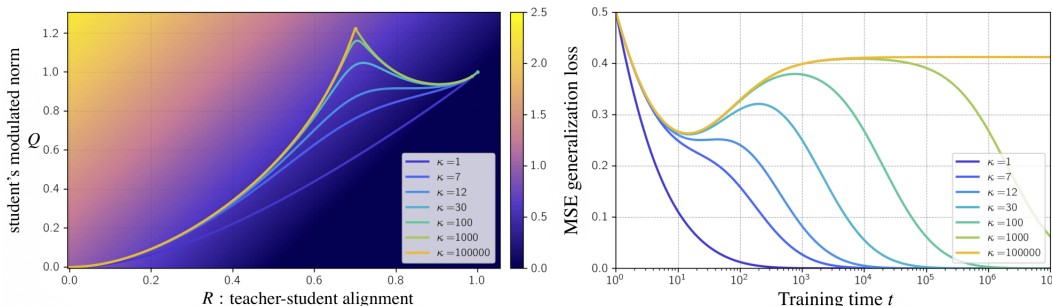

**Figure 3: Left**: Phase diagram of the generalization error as a function of $R(t)$ and $Q(t)$ (Eqs. 13 and 15). The generalization error for all pairs of $(R, Q) \in [0.0, 0.8] \times [0.0, 1.6]$ is contour-plotted in the background in shades of beige, with the best generalization performance being attained on the lower right part of the plot. The trajectories describe the evolution of $R(t)$ and $Q(t)$ as training proceeds. Each trajectory correspond to a different $\kappa$, the condition number of the modulation matrix $\boldsymbol{F}$ in Eq. 2. $\kappa$ describes the ratio of the rates at which two sets of features are learned. **Right**: The corresponding generalization curves for different plotted over the training time axis. **Analysis**: The trajectory with $\kappa = 1e5$ (bright yellow) starts at the origin and advances towards point $A$ (a descent in generalization error). Then by over-training, it converges to point $B$ (an ascent in generalization error). For the other trajectories with smaller $\kappa$, a first descent in generalization error occurs up to the point $A$, then an ascent happens, but they no longer converge to point $B$. Instead, by further training, these trajectories converge to point $C$ implying a second descent.

## 4 RELATED WORK AND DISCUSSION

Although the term *double descent* has been introduced rather recently (Belkin et al., 2019a), similar behaviors had already been observed and studied in several decades-old works form a statistical physics perspective (Krogh & Hertz, 1992a; Opper, 1995; Opper & Kinzel, 1996; Bös, 1998; Engel & Van den Broeck, 2001). More recently, these behaviors have been investigated in the context of modern machine learning, both from an empirical (Nakkiran et al., 2019; Amari et al., 2020; Yang et al., 2020) and theoretical (Belkin et al., 2019a; Geiger et al., 2019; Advani & Saxe, 2017; Mei & Montanari, 2019; Gerace et al., 2020; d'Ascoli et al., 2020; Ba et al., 2019; d'Ascoli et al., 2021) perspectives.

Hastie et al. (2019); Advani et al. (2020); Belkin et al. (2020) use random matrix theory (RMT) tools to characterize the asymptotic generalization behavior of over-parameterized linear and random feature models. In an influential work, Mei & Montanari (2019) extend the same analysis to a random feature model and theoretically derive the model-wise double descent curve for a model with Tikhonov regularization. Jacot et al. (2020) also study double descent in ridge estimators and show an equivalence to kernel ridge regression. Pennington & Worah (2019) used RMT to study the curvature of single-hidden-layer neural network in an attempt to understand the efficacy of first-order optimization methods in training DNNs. In addition, Liang & Rakhlin (2020) take a similar approach to investigate implicit regularization in high dimensional ridgeless regression with nonlinear kernels.

While most of the related work study the non-monotonicity of the generalization error as a function of the model size or sample size, Nakkiran et al. (2019) introduced the epoch-wise double descent. Epoch-wise double descent refers to the phenomenon where the generalization error undergoes two descents as the training time increases. There has been limited work on studying of epoch-wise double descent. Very recently, Heckel & Yilmaz (2020) and Stephenson & Lee (2021) have focused on finding the roots of this phenomenon.

Heckel & Yilmaz (2020) provides *upper bounds* on the risk of single and two layer models in a regression setting where the input data has distinct feature variances. Heckel & Yilmaz (2020) demonstrate that a superposition of two or more bias-variance tradeoff curves leads to epoch-wise double descent. The authors also show that different layers of the network are learned at different epochs. For that reason, epoch-wise double descent can be eliminated by appropriate selection of learning rates for individual network weights. Consistent with these findings, our work formalizes this phenomenon in terms of feature learning scales and provides closed-form predictions.

Stephenson & Lee (2021) arrives at similar conclusions. Authors in Stephenson & Lee (2021) take a random matrix theory approach on a data model that exhibits epoch-wise double descent. The data model is constructed so that the noise is explicitly added *only* to the fast-learning features while slow-learning features remain noise-free. Consequently, the fast-learning features are noisy and hence show a U-shaped generalization curve while slow-learning features are noiseless.

Our findings and those of Heckel & Yilmaz (2020) and Stephenson & Lee (2021) reinforce one another with a common central finding that the epoch-wise double descent results from different features/layers being learned at different time-scale. However, we also highlight that both Heckel & Yilmaz (2020) and Stephenson & Lee (2021) built upon tool from random matrix theory and study distinct data models from our teacher-student setup. We study the same phenomenon from a different perspective. By leveraging the replica method from statistical physics, we characterized the generalization behavior using a set of informative macroscopic parameters. While supporting the notion that the interaction of different feature learning speeds causes epoch-wise double descent, our work provides formal predictions of the dynamics that unfold during training.

We believe our theoretical framework sets the stage for further understanding of generalization dynamics in neural networks beyond the double descent. A future direction to study is a case in which the first descent is strong enough to bring down the training loss to very small values to the point that learning slower features is practically impossible or happens after a very large number of epochs. Power et al. (2021) reports an instance of such behavior called *Grokking* where the model abruptly learns to perfectly generalize but long after the training loss has reached very small values.

**Limitations.** It should be noted that studying finer details of the dynamics would require a more precise model of the neural networks. Clearly, our proposed model is not a universal and unique way to model the dynamics of the complex, over-parameterized deep neural networks.

**Social Impact.** The authors do not foresee a negative social impact specifically arising from this rather theoretical work.

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

## A  FURTHER RELATED WORK AND DISCUSSION

If we consider plots where the generalization error on the $y$-axis is plotted against other quantities on the $x$-axis, we find earlier works that have identified double descent behavior for quantities such as the number of parameters, the dimensionality of the data, the number of training samples, or the training time on the $x$-axis. In this paper, we studied epoch-wise double descent, *i.e.* we plot the training time $t$, or the number of training epochs, on the $x$-axis. Literature displaying double descent phenomena in generalization behavior w.r.t. other quantities do so in the limit of $t \to \infty$.

From a random matrix theory perspective, Le Cun et al. (1991); Hastie et al. (2019); Advani et al. (2020), and Belkin et al. (2020) are among works which have analytically studied the spectral density of the Hessian matrix. According to their analyses, at intermediate levels of complexity, the presence of small but non-zero eigenvalues in the Hessian matrix results in high generalization error as the inverse of the Hessian is calculated for the pseudo-inverse solution.

Neyshabur et al. (2014) demonstrated that over-parameterized networks does not necessarily overfit thus suggesting the need of a new form of measure of model complexity other than network size. Subsequently, Neyshabur et al. (2018) suggest a novel complexity measure based on unit-wise capacities which correlates better with the behavior of test error with increasing network size. Chizat & Bach (2020) study the global convergence and superior generalization behavior of infinitely wide two-layer neural networks with logistic loss. Goldt et al. (2020) make use of the Gaussian Equivalence Theorem to study the generalization performance of two-layer neural networks and kernel models trained on data drawn from pre-trained generative models. Bai & Lee (2020) investigated the gap between the empirical performance of over-parameterized networks and their NTK counterparts, first proposed by Jacot et al. (2018).

From the perspective of bias/variance trade-off, Geman et al. (1992), and more recently, Neal et al. (2018) empirically observe that while bias is monotonically decreasing, variance could be decreasing too or unimodal as the number of parameters increases, thus manifesting a double descent generalization curve. Hastie et al. (2019) analytically study the variance. More recently, Yang et al. (2020) provides a new bias/variance decomposition of bias exhibiting double desc-nt in which the variance follows a bell-shaped curve. However, the decrease in variance as the model size increases remains unexplained. For high dimensional regression with random features, d'Ascoli et al. (2020) provides an asymptotic expression for the bias/variance decomposition and identifies three sources of variance with non-monotonous behavior as the model size or dataset size varies. d'Ascoli et al. (2020) also employs the analysis of random feature models and identifies two forms of overfitting which leads to the so-called sample-wise triple descent. More recently, Chen et al. (2020) show that as a result of the interaction between the data and the model, one may design generalization curves with multiple descents.

From a statistical physics perspective, Opper (1995); Bös et al. (1993); Bös (1998); Opper & Kinzel (1996) are among the first studies which theoretically observe sample-wise double-descent in a ridge regression setup where the solution is obtained by the pseudo-inverse method. Most of these studies employ the "Gardner analysis" (Gardner, 1988; Gardner & Derrida, 1988; 1989) for models where the number of parameters and the dimensionality of data are coupled and hence the observed form of double descent is different from that observed in deep neural networks. A beautiful extended review of this line of work is provided in Engel & Van den Broeck (2001). Among recent works, Gerace et al. (2020) also apply the Gardner analysis but to a novel generalized data generating process called the hidden manifold model and derive the model-wise double-descent equations analytically.

Finally, recall that towards providing an explanation for the epoch-wise double descent, we argue that *the epoch-wise double descent can be attributed to different features being learned at different time-scales*, resulting in a non-monotonous generalization curve. In relation to the aspect of different feature learning scales, Rahaman et al. (2019) had observed that DNNs have a tendency towards learning simple target functions first that can allow for good generalization behavior of various data samples. Pezeshki et al. (2020) also identify and provide explanation for a feature learning imbalance exhibited by over-parameterized networks trained via gradient descent on cross-entropy loss, with the networks learning only a subset of the full feature spectrum over training. More recently though, Zhang & Wu (2020), show that certain DNNs models prioritize learning high-frequency components first followed by the learning of slow but informative features, leading to the second descent of the test error as observed in epoch-wise double descent.

**On the difference between model-wise and epoch-wise double descent curves.** In accordance with its name, model-wise double descent (in the test error) occurs due to an increase in model-size (number of its parameters), i.e., as the model transitions from an under-parameterized to an over-parameterized regime. A variety of works have tried to understand this phenomenon from the lens of implicit regularization (Neyshabur et al., 2014) or defining novel complexity measures (Neyshabur et al., 2017). On the other hand, epoch-wise double descent (in the test error) as treated in our work, is observed to occur for both over-parameterized (Nakkiran et al., 2019) and under-parameterized (Heckel & Yilmaz, 2020) setups. As found in our work along with the latter reference, this phenomenon seems to be a result of different feature learning speeds rather than the extent of model parameterization. The overlap of the test-error contributions from the different weights with varying scales of learning henceforth leads to a non-monotonous evolution of the model test error as exemplified by epoch-wise double descent.

We also note that the peak in model-wise double descent is associated with the model's capacity to perfectly interpolate the data, we do not think an analogous notion exists for the case of epoch-wise double descent. Our understanding of the peak in the latter is that it corresponds to a training time configuration whereby a subclass of features are already learnt (due to a larger associated signal-to-noise-ratio) and are being overfitted upon to fit the target. As training proceeds further, the remaining set of features are eventually learnt thus allowing for a lowering of the test error.

**On the implicit regularization of SGD and ridge-regularized loss.** The results presented in Eqs. 20-23 have a core dependence on the findings of Ali et al. (2019; 2020). These works first formalize the connection between (continuous-time) GD or SGD-based training of an ordinary least squares (OLS) setup and that of ridge regression, providing bounds on the test error under these algorithms

over training time $t$, in terms of a ridge setup with ridge parameter $\lambda = 1/t$. We utilize these results in the sense that by evaluating the generalization error $\mathcal{L}_G$ of our student-teacher setup with explicit ridge regularization, we invoke the connection between the ridge coefficient $\lambda$ and training time $t$ as described in these works, to obtain the behavior of (ridgeless) $\mathcal{L}_G$ over training. This determination of an expression of $\mathcal{L}_G(t)$ is what allows us to study the epoch-wise DD phenomenon.

## B    TECHNICAL PROOFS

### B.1    THE GENERALIZATION ERROR AS A FUNCTION OF $R$ AND $Q$ (EQ. 6)

Recall that the teacher is the data generator and is defined as,

$$y := y^* + \epsilon, \qquad y^* := \boldsymbol{z}^T W, \qquad z_i \sim \mathcal{N}(0, \frac{1}{\sqrt{d}}), \tag{26}$$

where $\boldsymbol{z} \in \mathbb{R}^d$ is the teacher's input and $y^*, y \in \mathbb{R}$ are the teacher's noiseless and noisy outputs, respectively. $W \in \mathbb{R}^d$ represents the (fixed) weights of the teacher and $\epsilon \in \mathbb{R}$ is the noise.

And student is defined as,

$$\hat{y} := \boldsymbol{x}^T \hat{W}, \qquad s.t. \qquad \boldsymbol{x} := F^T \boldsymbol{z}, \tag{27}$$

where the matrix $F \in \mathbb{R}^{d \times d}$ is a predefined and fixed modulation matrix regulating the student's access to the true input $\boldsymbol{z}$.

The average generalization error of the student, determined by averaging the student's error over all possible input-target pairs and noise realizations is given by,

$$\mathcal{L}_G := \frac{1}{2} \mathbb{E}_{\boldsymbol{x}, W} \big[ (y^* - \hat{y})^2 \big], \tag{28}$$

in which the variables $(y^*, \hat{y})$ form a bi-variate Gaussian distribution with zero mean and a covariance of,

$$\Sigma = \begin{bmatrix} < y^*, y^* >_z & < y^*, \hat{y} >_z \\ < y^*, \hat{y} >_z & < \hat{y}, \hat{y} >_z \end{bmatrix} = \begin{bmatrix} 1 & R \\ R & Q \end{bmatrix}, \tag{29}$$

in which,

$$R := \mathbb{E}_z[y^{*T} \hat{y}] = \mathbb{E}_z[W^T z z^T F \hat{W}] = \frac{1}{d} W^T F \hat{W}, \quad \text{and,} \tag{30}$$

$$Q := \mathbb{E}_z[\hat{y}^T \hat{y}] = \mathbb{E}_z[\hat{W}^T F^T z z^T F \hat{W}] = \frac{1}{d} \hat{W}^T F^T F \hat{W}. \tag{31}$$

Eq. 29 implies a correlation between $y^*$ and $\hat{y}$ obstructing the calculation of the average in Eq. 28. Following (Bös, 1998; Krogh & Hertz, 1992a), we define decoupled variables $\tilde{y}^*$ and $\tilde{\hat{y}}$ as follows,

$$y^* =: \tilde{y}^*, \quad \text{and} \quad \hat{y} =: R\tilde{y}^* + \sqrt{Q - R^2}\tilde{\hat{y}}. \tag{32}$$

The variables $\tilde{y}^*$ and $\tilde{\hat{y}}$ are independent Gaussian variables such that $< \tilde{y}^*, \tilde{\hat{y}} >_z = 0$. Therefore, two expectations can be applied independently,

$$\mathcal{L}_G := \frac{1}{2} \mathbb{E}_{\boldsymbol{x}, W} \big[ (y^* - \hat{y})^2 \big], \tag{33}$$

$$= \frac{1}{2} \mathbb{E}_{\tilde{y}^*, \tilde{\hat{y}}} \big[ (\tilde{y}^* - (R\tilde{y}^* + \sqrt{Q - R^2}\tilde{\hat{y}}))^2 \big], \tag{34}$$

$$= \frac{1}{2} (1 + Q - 2R). \tag{35}$$

Finally, we note that expectation w.r.t. a Gaussian variable $x$ is defined as,

$$\mathbb{E}_x[f(x)] := \int_{-\infty}^{+\infty} \frac{dx}{\sqrt{2\pi}} \exp\big(-\frac{x^2}{2}\big) f(x). \tag{36}$$

## B.2 THE GENERAL CASE EXACT DYNAMICS (EQS. 9-10)

Recall that to train our student network, we use stochastic gradient descent (SGD) on the regularized mean squared loss, evaluated on the $n$ training examples as,

$$\mathcal{L}_{\mathcal{T}} := \frac{1}{2n} \sum_{\mu=1}^{n} (y^{\mu} - \hat{y}^{\mu})^2 + \frac{\lambda}{2} ||\hat{W}||_2^2, \tag{37}$$

where $\lambda \in [0, \infty)$ is the regularization coefficient.

The minimum of the loss function, denoted by $\overline{W}$, is achieved at,

$$\nabla_{\hat{W}} \mathcal{L}_{\mathcal{T}} = 0 \Rightarrow \nabla_{\hat{W}} \left[ \frac{1}{2} ||y - X\hat{W}||_2^2 + \frac{\lambda}{2} ||\hat{W}||_2^2 \right] = 0 \tag{38}$$

$$\Rightarrow -X^T(y - X\hat{W}) + \lambda\hat{W} = 0 \tag{39}$$

$$\Rightarrow \overline{W} := (X^T X + \lambda I)^{-1} X^T y. \tag{40}$$

An exact gradient descent has the following dynamics,

$$\hat{W}_t = \hat{W}_{t-1} - \eta \nabla_{\hat{W}_{t-1}} \mathcal{L}_{\mathcal{T}}, \tag{41}$$

$$= \hat{W}_{t-1} - \eta \left[ -X^T(y - X\hat{W}_{t-1}) + \lambda\hat{W}_{t-1} \right] \tag{42}$$

$$= (1 - \eta\lambda)\hat{W}_{t-1} - \eta X^T X \hat{W}_{t-1} + \eta X^T y, \tag{43}$$

$$= [(1 - \eta\lambda)I - \eta X^T X]\hat{W}_{t-1} + \eta X^T y, \tag{44}$$

$$= [(1 - \eta\lambda)I - \eta X^T X]\hat{W}_{t-1} + \eta(X^T X + \lambda I)(X^T X + \lambda I)^{-1} X^T y, \tag{45}$$

$$= [(1 - \eta\lambda)I - \eta X^T X]\hat{W}_{t-1} + \eta(X^T X + \lambda I)\overline{W}, \tag{46}$$

$$= [(1 - \eta\lambda)I - \eta X^T X]\hat{W}_{t-1} + (\eta X^T X + \eta\lambda I)\overline{W}, \tag{47}$$

$$= [(1 - \eta\lambda)I - \eta X^T X]\hat{W}_{t-1} + (\eta X^T X + (\eta\lambda - 1)I)\overline{W} + \overline{W}, \tag{48}$$

which leads to,

$$\hat{W}_t - \overline{W} = [(1 - \eta\lambda)I - \eta X^T X](\hat{W}_{t-1} - \overline{W}), \tag{49}$$

$$= [(1 - \eta\lambda)I - \eta X^T X]^t(\hat{W}_0 - \overline{W}). \tag{50}$$

Assuming $\hat{W}_0 = 0$, we arrive at the following closed-form equation,

$$\hat{W}_t = \left( I - [(1 - \eta\lambda)I - \eta X^T X]^t \right) \overline{W}, \tag{51}$$

where $\overline{W}$ is defined in Eq 40.

Now back to definition of $R$ in Eq. 84 and by substitution of Eq. 51, we have,

$$R(t) := \frac{1}{d} W^T F \hat{W}_t, \tag{52}$$

$$= \frac{1}{d} W^T F \left( I - [(1 - \eta\lambda)I - \eta X^T X]^t \right) \overline{W}, \tag{53}$$

$$= \frac{1}{d} W^T F \left( I - [(1 - \eta\lambda)I - \eta X^T X]^t \right) (X^T X + \lambda I)^{-1} X^T y, \tag{54}$$

$$= \frac{1}{d} W^T F V \left( I - [(1 - \eta\lambda)I - \eta\Lambda]^t \right) (\Lambda + \lambda I)^{-1} V^T X^T y, \quad (X^T X = V\Lambda V^T) \tag{55}$$

$$= \frac{1}{d} W^T F V \left( I - [(1 - \eta\lambda)I - \eta\Lambda]^t \right) (\Lambda + \lambda I)^{-1} (\Lambda V^T F^{-1} W + \Lambda^{\frac{1}{2}} \epsilon), \tag{56}$$

$$= \frac{1}{d} W^T F V \left( I - [(1 - \eta\lambda)I - \eta\Lambda]^t \right) (\Lambda + \lambda I)^{-1} (\Lambda V^T F^{-1} W + \Lambda^{\frac{1}{2}} \epsilon), \tag{57}$$

$$= \frac{1}{d} \mathbf{Tr} \left[ \left( I - [(1 - \eta\lambda)I - \eta\Lambda]^t \right) \frac{\Lambda}{\Lambda + \lambda I} \right]. \tag{58}$$

Similarly for $Q$, let $D := \left( I - \left[ (1 - \eta\lambda)I - \eta\Lambda \right]^t \right)$, then we have,

$$Q(t) := \frac{1}{d}\hat{W}^T F^T F \hat{W}, \tag{59}$$

$$= \frac{1}{d}\overline{W}^T \left( I - \left[ (1-\eta\lambda)I - \eta X^T X \right]^t \right) F^T F \left( I - \left[ (1-\eta\lambda)I - \eta X^T X \right]^t \right)\overline{W}, \tag{60}$$

$$= \frac{1}{d}\overline{W}^T V D V^T F^T F V D V^T \overline{W}, \tag{61}$$

$$= \frac{1}{d}\overline{W}^T V D \tilde{F}^T \tilde{F} D V^T \overline{W}, \qquad (\tilde{F} := FV, X = U\Lambda^{1/2}V^T, \tilde{\epsilon} := U^T \epsilon) \tag{62}$$

$$= \frac{1}{d}(W^T F^{-1^T} V + \Lambda^{-1/2}\tilde{\epsilon})\frac{\Lambda}{\Lambda + \lambda I} D \tilde{F}^T \tilde{F} D \frac{\Lambda}{\Lambda + \lambda I}(V^T F^{-1} W + \Lambda^{-1/2}\tilde{\epsilon}), \tag{63}$$

$$= \frac{1}{d}(W^T \tilde{F}^{-1^T} + \Lambda^{-1/2}\tilde{\epsilon})\frac{\Lambda}{\Lambda + \lambda I} D \tilde{F}^T \tilde{F} D \frac{\Lambda}{\Lambda + \lambda I}(\tilde{F}^{-1} W + \Lambda^{-1/2}\tilde{\epsilon}), \tag{64}$$

$$= \frac{1}{d}W^T \tilde{F}^{-1^T} \frac{\Lambda}{\Lambda + \lambda I} D \tilde{F}^T \tilde{F} D \frac{\Lambda}{\Lambda + \lambda I}\tilde{F}^{-1} W, \tag{65}$$

$$+ \frac{1}{d}\Lambda^{-1/2}\tilde{\epsilon}\frac{\Lambda}{\Lambda + \lambda I} D \tilde{F}^T \tilde{F} D \frac{\Lambda}{\Lambda + \lambda I}\Lambda^{-1/2}\tilde{\epsilon}, \tag{66}$$

$$= \frac{1}{d}\mathbf{Tr}\left[ A^T A \right] + \frac{\sigma_\epsilon^2}{d}\mathbf{Tr}\left[ B^T B \right] \tag{67}$$

where,

$$A := \tilde{F}\left( I - \left[ (1-\eta\lambda)I - \eta\Lambda \right]^t \right)\frac{\Lambda}{\Lambda + \lambda I}\tilde{F}^{-1} \quad \text{and,} \tag{68}$$

$$B := \tilde{F}\left( I - \left[ (1-\eta\lambda)I - \eta\Lambda \right]^t \right)\frac{\Lambda}{\Lambda + \lambda I}\Lambda^{-\frac{1}{2}}. \tag{69}$$

For simplicity and brevity of the results, in the main text, we only present the results where $\sigma_\epsilon^2 = 0$ and $\lambda = 0$. Substituting $\sigma_\epsilon^2 = \lambda = 0$ leads to the following expressions,

$$R(t) = \frac{1}{d}\mathbf{Tr}\left[ \left( I - I - \eta\Lambda \right]^t \right) \right]. \tag{70}$$

$$Q(t) = \frac{1}{d}\mathbf{Tr}\left[ A^T A \right] \quad \text{where,} \quad A := FV\left( I - \left[ I - \eta\Lambda \right]^t \right)V^T F^{-1}, \tag{71}$$

and that concludes the proof.

## B.3 THE SPECIAL CASE APPROXIMATE DYNAMICS (EQS. 13 AND 15)

Recall that the teacher and student are defined as,

$$y := y^* + \epsilon, \qquad y^* := z^T W, \qquad \hat{y} := x^T \hat{W}, \qquad x := F^T z, \tag{72}$$

where $\epsilon \sim \mathcal{N}(0, \sigma_\epsilon^2)$ is the label noise, $F$ is the modulation matrix, and $||z||_2^2 = ||W||_2^2 = 1$.

The training and generalization losses are defined as,

$$\mathcal{L}_T := \frac{1}{2n}\sum(\hat{y} - y)^2 + \frac{\lambda}{2}||\hat{W}||_2^2, \qquad \mathcal{L}_G := \frac{1}{2}\mathbb{E}_z[(\hat{y} - y^*)^2]. \tag{73}$$

According to Eq. 6, the generalization loss can be written in terms of two scalar variables $R$ and $Q$,

$$\mathcal{L}_G = \frac{1}{2}(1 + Q - 2R), \quad \text{where,} \tag{74}$$

$$R := \mathbb{E}_z[y^{*T}\hat{y}] = \mathbb{E}_z[W^T z z^T F \hat{W}] = \frac{1}{d}W^T F \hat{W}, \quad \text{and,} \tag{75}$$

$$Q := \mathbb{E}_z[\hat{y}^T \hat{y}] = \mathbb{E}_z[\hat{W}^T F^T z z^T F \hat{W}] = \frac{1}{d}\hat{W}^T F^T F \hat{W}. \tag{76}$$

Now, applying $t$ steps of SGD on $\mathcal{L}_T$ results in the following distribution for the student's weights,

$$P(\hat{W}, t) = \frac{1}{Z_{\beta,t}} e^{-\beta \tilde{\mathcal{L}}_T(\hat{W},t)}, \tag{77}$$

in which $\tilde{\mathcal{L}}_T(\hat{W}, t)$ is a modified loss where its equilibrium coincides with the $t^{th}$ iterate of SGD on the original loss $\mathcal{L}_T(\hat{W})$.

In Eq. 77 the scalar variable $\beta$ depends on the noise of SGD and $Z_{\beta,t}$ is the partition function which is defined as,

$$Z_{\beta,t} = \frac{\int_{-\infty}^{\infty} \prod_{i=1}^{d} \mathrm{d}(\hat{W}_i) \delta\left(\frac{1}{d}\hat{W}_i^T F^T F \hat{W}_i - Q_0\right) P(\hat{W}_i, t)}{\int_{-\infty}^{\infty} \prod_{i=1}^{d} \mathrm{d}(\hat{W}_i) \delta\left(\frac{1}{d}\hat{W}_i^T F^T F \hat{W}_i - Q_0\right)}, \tag{78}$$

in which, $Q_0$ can be perceived to be a target norm the student weights $\hat{W}$ are being constrained to and $d$ is the dimensionality of the data. It can be interpreted that the partition function $Z_{\beta,t}$ counts the students.

We are now interested in finding $R$ and $Q$ of the typical (most probable) students. Therefore, it suffices to find the students that dominate the partition function (or more precisely the free-energy). The free-energy is defined as,

$$f := -\frac{1}{\beta d} \mathbb{E}_{W,z}\big[\ln Z_{\beta,t}\big], \tag{79}$$

where $W$ and $z$ are the teacher's weight and input, respectively.

Due to the logarithm inside the expectation, analytical computation of Eq. 79 is intractable. However, the replica method (Mézard et al., 1987) allows us to tackle this through the following identity,

$$\mathbb{E}_{W,z}[\ln Z_{\beta,t}] = \lim_{r \to 0} \frac{\mathbb{E}_{W,z}[Z_{\beta,t}^r] - 1}{r}. \tag{80}$$

**The case where $F = I$.** As a first step, we first study a case where $F = I$. In that case, as derived in Bös (1998), Eq. 79 can be simplified to,

$$-\beta f = \frac{1}{2}\frac{Q - R^2}{Q_0 - Q} + \frac{1}{2}\ln(Q_0 - Q) - \frac{n}{2d}\ln[1 + \beta(Q_0 - Q)] - \frac{n\beta}{2d}\frac{G - 2HR + Q}{1 + \beta(Q_0 - Q)}, \tag{81}$$

in which the scalar variables $G$ and $H$ are defined as,

$$H := \mathbb{E}_{y^*}[y^{*T} y] = \mathbb{E}_{y^*}[y^{*T}(y^* + \epsilon)] = 1, \tag{82}$$

$$G := \mathbb{E}_{y^*}[y^T y] = \mathbb{E}_{y^*}[(y^* + \epsilon)^T(y^* + \epsilon)] = 1 + \sigma_\epsilon^2. \tag{83}$$

At this point, in order to find the most probable students, one can extremize the free-energy $f(R, Q, Q_0)$ in Eq. 81. The solution to this extermination is derived in Bös et al. (1993) and reads,

$$\nabla_R f = 0 \quad \Rightarrow \quad R = \frac{n}{d}\frac{1}{a}, \tag{84}$$

$$\nabla_Q f = 0 \quad \Rightarrow \quad Q = \frac{n}{d}\frac{1}{a^2 - n/d}\left(G - \frac{n}{d}\frac{2 - a}{a}\right), \tag{85}$$

$$\nabla_{Q_0} f = 0 \quad \Rightarrow \quad a = 1 + \frac{2\tilde{\lambda}}{1 - n/d - \tilde{\lambda} + \sqrt{(1 - n/d - \tilde{\lambda})^2 + 4\tilde{\lambda}}}, \tag{86}$$

in which,

$$a := 1 + \frac{1}{\beta(Q_0 - Q)}, \quad \text{and,} \quad \tilde{\lambda} := \lambda + \frac{1}{\eta t}. \tag{87}$$

**The case where $F$ follows Assumption 1.**

**Assumption.** *The modulation matrix, $F$, under a SVD, $F := U\Sigma V^T$ has two sets of singular values such that the first $p$ singular values are equal to $\sigma_1$ and the remaining $d-p$ singular values are equal to $\sigma_2$. We let the condition number of $F$ to be denoted by $\kappa := \frac{\sigma_1}{\sigma_2} > 1$.*

Without loss of generality, we assume that $U = V = I$. Consequently, the (noiseless) teacher and the student can be written as the composition of two sub-models as following,

$$y^* = y_1^* + y_2^* = z_1^T W_1 + z_2^T W_2, \qquad \text{(teacher decomposition)} \qquad (88)$$

$$\hat{y} = \hat{y}_1 + \hat{y}_2 = \sigma_1 z_1^T \hat{W}_1 + \sigma_2 z_2^T \hat{W}_2, \qquad \text{(student decomposition)} \qquad (89)$$

in which $z_1 \in \mathbb{R}^p$ and $z_2 \in \mathbb{R}^{d-p}$.

Let $\hat{y}_i$ denote the output of the $i^{th}$ component of the student. Also let $y_i^*$ and $y_i$ denote the noiseless and noisy targets, respectively. Therefore, for the student components $i \in 1, 2$, we have,

$$\hat{y}_1 = \sigma_1 z_1^T \hat{W}_1,$$
$$y_1^* = z_1^T W_1,$$
$$y_1 = y_1^* + \underbrace{z_2^T W_2 - \sigma_2 z_2^T \hat{W}_2}_{y_2^* - \hat{y}_2 = \epsilon_2(t)} + \epsilon,$$

$$\hat{y}_2 = \sigma_2 z_2^T \hat{W}_2,$$
$$y_2^* = z_2^T W_2,$$
$$y_2 = y_2^* + \underbrace{z_1^T W_1 - \sigma_1 z_1^T \hat{W}_1}_{y_1^* - \hat{y}_1 = \epsilon_1(t)} + \epsilon,$$

in which $\epsilon$ is the *explicit noise*, added to the teacher's output while $\epsilon_j(t)$ is an *implicit variable noise* which decreases as the component $j \neq i$ learns to match $\hat{y}_j$ and $y_j$.

Accordingly, the variables $H_i$ and $G_i$ for each component $i$ are re-defined as,

$$H_1 = \mathbb{E}[y_1^{*T} y_1] = \mathbb{E}_{y_1^*}[y_1^{*T} y_1^*] = \frac{p}{d},$$
$$G_1 = \mathbb{E}[y_1^T y_1],$$
$$= \mathbb{E}[(y_1^* + y_2^* - \hat{y}_2)^T (y_1^* + y_2^* - \hat{y}_2)] + \sigma_\epsilon^2,$$
$$= \mathbb{E}[y_1^{*T} y_1^*] + \mathbb{E}[y_2^{*T} y_2^*] + \mathbb{E}[\hat{y}_2^T \hat{y}_2],$$
$$\quad - 2\mathbb{E}[y_2^{*T} \hat{y}_2] + \sigma_\epsilon^2,$$
$$= \frac{p}{d} + \frac{d-p}{d} + Q_2 - 2R_2 + \sigma_\epsilon^2,$$
$$= 1 + Q_2 - 2R_2 + \sigma_\epsilon^2,$$

$$H_2 = \mathbb{E}[y_2^{*T} y_2] = \mathbb{E}_{y_2^*}[y_2^{*T} y_2^*] = \frac{d-p}{d},$$
$$G_2 = \mathbb{E}[y_2^T y_2],$$
$$= \mathbb{E}[(y_2^* + y_1^* - \hat{y}_1)^T (y_2^* + y_1^* - \hat{y}_1)] + \sigma_\epsilon^2,$$
$$= \mathbb{E}[y_2^{*T} y_2^*] + \mathbb{E}[y_1^{*T} y_1^*] + \mathbb{E}[\hat{y}_1^T \hat{y}_1],$$
$$\quad - 2\mathbb{E}[y_1^{*T} \hat{y}_1] + \sigma_\epsilon^2,$$
$$= \frac{d-p}{d} + \frac{p}{d} + Q_1 - 2R_1 + \sigma_\epsilon^2,$$
$$= 1 + Q_1 - 2R_1 + \sigma_\epsilon^2,$$

in which $R_i$ and $Q_i$ are defined as,

$$R_i := \mathbb{E}_z[y_i^{*T} \hat{y}_i] = \frac{1}{d} W_i^T \sigma_i \hat{W}_i, \quad \text{and,} \quad Q_i := \mathbb{E}_z[\hat{y}_i^T \hat{y}_i] = \frac{1}{d} \hat{W}_i^T \sigma_i^2 \hat{W}_i,$$

where $\sigma_i$ denotes the singular values of the matrix $F$ as defined in Assumption 1.

Rewriting Eqs. 84, 85, and 86 for each of the student's components, we arrive at,

$$R_1 = \frac{n}{d} \frac{1}{a_1},$$
$$Q_1 = \frac{n}{pa_1^2 - n} \left( 1 + Q_2 - 2R_2 + \sigma_\epsilon^2 - \frac{n}{d} \frac{2 - a_1}{a_1} \right),$$
$$a_1 = 1 + \frac{2\tilde{\lambda}_1}{1 - \frac{n}{p} - \tilde{\lambda}_1 + \sqrt{(1 - \frac{n}{p} - \tilde{\lambda}_1)^2 + 4\tilde{\lambda}_1}},$$
$$\tilde{\lambda}_1 := \frac{d}{p} \frac{1}{\sigma_1^2} (\lambda + \frac{1}{\eta t}),$$

$$R_2 = \frac{n}{d} \frac{1}{a_2},$$
$$Q_2 = \frac{n}{(d-p)a_1^2 - n} \left( 1 + Q_1 - 2R_1 + \sigma_\epsilon^2 - \frac{n}{d} \frac{2 - a_2}{a_2} \right),$$
$$a_2 = 1 + \frac{2\tilde{\lambda}}{1 - \frac{n}{d-p} - \tilde{\lambda} + \sqrt{(1 - \frac{n}{d-p} - \tilde{\lambda})^2 + 4\tilde{\lambda}}},$$
$$\tilde{\lambda}_2 := \frac{d}{d-p} \frac{1}{\sigma_2^2} (\lambda + \frac{1}{\eta t}),$$

where $Q_1$ depends on $Q_2$ and vice versa. However, with simple calculations, we can arrive at the following standalone equation. Let,

$$\alpha_1 = \frac{n}{p}, \ \alpha_2 = \frac{n}{d-p}, \tag{90}$$

and also let,

$$b_i = \frac{\alpha_i}{a_i^2 - \alpha_i}, \quad c_i = 1 - 2R_i - \frac{n}{d}\frac{2 - a_i}{a_i} \quad \text{for} \quad i \in \{1, 2\}, \tag{91}$$

with which the closed-from scalar expression for $Q(t, \lambda)$ reads,

$$Q(t, \lambda) = Q_1 + Q_2, \quad \text{where,} \quad Q_1 := \frac{b_1 b_2 c_2 + b_1 c_1}{1 - b_1 b_2}, \quad \text{and,} \quad Q_2 := \frac{b_1 b_2 c_1 + b_2 c_2}{1 - b_1 b_2}. \tag{92}$$

### B.4 REPLICA TRICK

In the following, we detail the mathematical arguments leading to the *replica trick* expression. For some $r \to 0$, we can write for any scalar $x$:

$$x^r = \exp(r \ln x) = \lim_{r \to 0} 1 + r \ln x$$
$$\Rightarrow \lim_{r \to 0} r \ln x = \lim_{r \to 0} x^r - 1$$
$$\Rightarrow \ln x = \lim_{r \to 0} \frac{x^r - 1}{r} \tag{93}$$
$$\therefore \mathbb{E}[\ln x] = \lim_{r \to 0} \frac{\mathbb{E}[x^r] - 1}{r}, \quad \mathbb{E} : \text{averaging}$$

### B.5 COMPUTATION OF THE FREE-ENERGY

The self-averaged free energy (per unit weight) of our student network, is given by (Engel & Van den Broeck, 2001),

$$-\beta f = \frac{1}{d} \langle\langle \ln Z \rangle\rangle_{z,W} \tag{94}$$

Here, $\beta = 1/T$ is the inverse temperature parameter corresponding to our statistical ensemble, $d$ the (teacher) student network width, and $Z$ the partition function of the system defined as ($n$: number of training examples).

As Gaussian variables (with $n, d \to \infty$), in the partition function, to obtain,

$$
\begin{aligned}
\langle\langle Z^r \rangle\rangle_{z,W} &= \prod_{a=1}^{r}\prod_{\mu=1}^{d} \int \mathrm{d}\mu\left(W^a\right) \mathrm{d}y_a^\mu \mathrm{d}(y^*)^\mu e^{-\beta N \mathcal{E}_\mathcal{T}(y_a, y^*)} \\
&\quad \times \left\langle\left\langle \delta\left(y^{*\mu} - \frac{1}{\sqrt{d}}W^T x^{*\mu}\right) \delta\left(y_a^\mu - \frac{1}{\sqrt{d}}W_a^T x^\mu\right)\right\rangle\right\rangle_{z,W} \\
&= \prod_{a=1}^{r}\prod_{\mu=1}^{d} \int \mathrm{d}\mu\left(W^a\right) \frac{\mathrm{d}y_a^\mu \mathrm{d}\hat{y}_a^\mu}{2\pi} \frac{\mathrm{d}y^{*\mu}\mathrm{d}\hat{y}^{*\mu}}{2\pi} e^{-\beta N \mathcal{E}_\mathcal{T}(y_a, y^*)} e^{iy^{*\mu}\hat{y}^{*\mu} + iy_a^\mu \hat{y}_a^\mu} \\
&\quad \times \left\langle\left\langle \exp\left(-\frac{i}{\sqrt{d}}\hat{y}^{*\mu}W^T x^{*\mu} - \frac{i}{\sqrt{d}}\hat{y}_a^\mu W_a^T x^\mu\right)\right\rangle\right\rangle_{z,W}
\end{aligned}
\tag{95}
$$

where in the last line above, we have expressed the inserted $\delta$ functions using their integral representations. To make further progress, we introduce the auxiliary variables,

$$\sum_{ija} W_a^i \Delta_{ij} W^{*j} = dR_a, \tag{96}$$

$$\sum_{ij\langle a,b \rangle} W_a^i \Gamma_{ij} W_b^j = dQ_{ab} \tag{97}$$

via the respective $\delta$ functions, to arrive at,

$$\langle\langle Z^n \rangle\rangle_{z,W} = \prod_{\mu,a,b} \int \mathrm{d}\mu\left(\mathbf{W}^a\right) \frac{\mathrm{d}y_a^\mu \mathrm{d}\hat{y}_a^\mu}{2\pi} \frac{\mathrm{d}y^{*\mu} \mathrm{d}\hat{y}^{*\mu}}{2\pi} e^{-\beta N \mathcal{E}_\mathcal{T}(y_a, y^*)} e^{iy^{*\mu}\hat{y}^{*\mu} + iy_a^\mu \hat{y}_a^\mu}$$

$$\times \int P \mathrm{d}Q^{ab} \int P \mathrm{d}R^a \, \delta\left(\sum_{i,j,a} W_a^i \Delta_{i,j} W^{*j} - PR_a\right) \delta\left(\sum_{ij\langle a,b\rangle} W_a^i \Gamma_{ij} W_b^j - PQ_{ab}\right)$$

$$\times \left\langle\left\langle \exp\left(-\frac{Q_0}{2}\sum_{\mu,a}(\hat{y}_a^\mu)^2 - \frac{1}{2}\sum_{\mu,\langle a,b\rangle}\hat{y}_a^\mu \hat{y}_b^\mu Q_{ab} - \sum_{\mu,a}\hat{y}^{*\mu}\hat{y}_a^\mu R_a - \frac{1}{2}\sum_\mu (\hat{y}^{*\mu})^2\right)\right\rangle\right\rangle_W$$
$$(98)$$

Repeating the procedure of expressing the above $\delta$ functions using their integral representations, we then get ($\alpha = n/d$),

$$\langle\langle Z^n \rangle\rangle_{x,x^*,W} = \int \prod_{a,b} \frac{\mathrm{d}Q_0}{\sqrt{2\pi}} \frac{\mathrm{d}\hat{Q}_{0a}}{4\pi} \frac{\mathrm{d}Q_{ab}\hat{Q}_{ab}}{2\pi/d} \frac{\mathrm{d}R_a \hat{R}_a}{2\pi/d} \exp\left(\frac{iP}{2}\sum_a Q_0 \hat{Q}_{0a} + iP\sum_{a<b} Q^{ab}\hat{Q}^{ab}\right.$$

$$+ iP\sum_a R^a \hat{R}^a\right) \int \prod_{i,a} \frac{\mathrm{d}W_i^a}{\sqrt{2\pi}} \exp\left(-\frac{i}{2}\sum_{i,j,a}\hat{Q}_{0a} W_a^i \Gamma_{ij} W_a^j\right.$$

$$- i\sum_{i,j,a<b}\hat{Q}_{ab} W_a^i \Gamma_{ij} W_b^j - i\sum_{i,j,a}\hat{R}_a \Delta_{ij} W_a^j\right) \times$$

$$\int \prod_{\mu,a} \frac{\mathrm{d}y_a^\mu \mathrm{d}\hat{y}_a^\mu}{2\pi} \frac{\mathrm{d}y^{*\mu}}{\sqrt{2\pi}} e^{-\beta N \mathcal{E}_\mathcal{T}(y_a, y^*)} \exp\left(-\frac{1}{2}\sum_\mu (y^{*\mu})^2 + i\sum_{\mu,a}\hat{y}_a^\mu \hat{y}_a^\mu\right.$$

$$\left.- \frac{1}{2}\sum_{a,\mu}\left(1 - R_a^2\right)(\hat{y}_a^\mu)^2 - \frac{1}{2}\sum_{\mu,\langle a,b\rangle}\hat{y}_a^\mu \hat{y}_b^\mu \left(Q^{ab} - R^a R^b\right) - i\sum_{\mu,a} y^{*\mu}\hat{y}_a^\mu R^a\right)$$
$$(99)$$

If we now, perform a singular value decomposition of the covariance matrix $\Gamma$ as, $\Gamma = \mathrm{U}^T \mathrm{S}\mathrm{U} = \mathrm{V}^T \mathrm{V}$, where $\mathrm{S}$: matrix of singular values of $\Gamma$, and we have expressed, $\mathrm{V} = \mathrm{S}^{1/2}\mathrm{U}$, then one can proceed to write,

$$\langle\langle Z^n \rangle\rangle_{x,W} = \frac{1}{\det|V|} \int \prod_{a,b} \frac{\mathrm{d}Q_0}{\sqrt{2\pi}} \frac{\mathrm{d}\hat{Q}_{0a}}{4\pi} \frac{\mathrm{d}Q_{ab}\hat{Q}_{ab}}{2\pi/d} \frac{\mathrm{d}R_a \hat{R}_a}{2\pi/d} \exp\left(\frac{iP}{2}\sum_a Q_0 \hat{Q}_{0a}\right.$$

$$+ iP\sum_{a<b} Q^{ab}\hat{Q}^{ab} + iP\sum_a R^a \hat{R}^a\right) \int \prod_{i,a} \frac{\mathrm{d}\tilde{W}_i^a}{\sqrt{2\pi}} \exp\left(-\frac{i}{2}\sum_{i,a}\hat{Q}_{0a}\left(\tilde{W}_a^i\right)^2\right.$$

$$- i\sum_{i,a<b}\hat{Q}_{ab}\tilde{W}_a^i \tilde{W}_b^i - i\sum_{i,j,a}\hat{R}_a \tilde{W}_a^j\right) \times \int \prod_{\mu,a} \frac{\mathrm{d}y_a^\mu \mathrm{d}\hat{y}_a^\mu}{2\pi} \frac{\mathrm{d}y^{*\mu}}{\sqrt{2\pi}} e^{-\beta N \mathcal{E}_\mathcal{T}(y_a, y^*)}$$

$$\exp\left(-\frac{1}{2}\sum_\mu (y_\mu^*)^2 + i\sum_{\mu,a}\hat{y}_a^\mu \hat{y}_a^\mu - \frac{1}{2}\sum_{a,\mu}\left(1 - R_a^2\right)(\hat{y}_a^\mu)^2 - i\sum_{\mu,a} y^{*\mu}\hat{y}_a^\mu R^a\right.$$

$$\left.- \frac{1}{2}\sum_{\mu,\langle a,b\rangle}\hat{y}_a^\mu \hat{y}_b^\mu \left(Q^{ab} - R^a R^b\right)\right)$$
$$(100)$$

having expressed, $\tilde{W}_a = VW_a$, and identifying $\Delta = \mathrm{S}^{1/2}\mathrm{U}$ from our definitions. Now, since in the above, the $W_i^a$ integrals factorize in $i$, and similarly the $y_a^\mu$, $\hat{y}_a^\mu$ and $\mathrm{d}y^{*\mu}$ factorize in $\mu$, one can proceed to write:

$$\langle\langle Z^n \rangle\rangle_{x,W} = \frac{1}{\det|V|} \int \prod_{a,b} \frac{\mathrm{d}Q_0 \mathrm{d}\hat{Q}_{0a}}{\sqrt{2\pi}4\pi} \frac{\mathrm{d}Q_{ab}\hat{Q}_{ab}}{2\pi/d} \frac{\mathrm{d}R_a \hat{R}_a}{2\pi/d} \exp\left(P\left[\frac{i}{2}\sum_a Q_0 \hat{Q}_{0a}\right.\right.$$

$$\left.\left.+ i\sum_{a<b} Q^{ab}\hat{Q}^{ab} + i\sum_a R^a \hat{R}^a + G_S(\hat{Q}_{0a}, \hat{Q}^{ab}, \hat{R}^a) + \alpha G_E(Q^{ab}, R^a)\right]\right)$$
$$(101)$$

where,

$$G_S(\hat{Q}_{0a}, \hat{Q}^{ab}, \hat{R}^a) = \ln \int \prod_a \frac{d\tilde{W}^a}{\sqrt{2\pi}} \exp\left(-\frac{i}{2}\sum_a \hat{Q}_{0a}\tilde{W}_a^i\tilde{W}_a^i - i\sum_{a<b}\hat{Q}_{ab}\tilde{W}_a\tilde{W}_b - i\sum_a \hat{R}_a\tilde{W}_a\right)$$

$$G_E(Q^{ab}, R^a) = \ln \int \prod_a \frac{dy_a d\hat{y}_a}{2\pi} \frac{dy^*}{\sqrt{2\pi}} e^{-\beta N \mathcal{E}_\mathcal{T}(y_a, y^*)} \exp\left(-\frac{1}{2}(y^*)^2 + i\sum_a \hat{y}_a\hat{y}_a\right.$$

$$\left. - \frac{1}{2}\sum_a \left(1 - R_a^2\right)(\hat{y}_a)^2 - \frac{1}{2}\sum_{\langle a,b\rangle} \hat{y}_a\hat{y}_b\left(Q^{ab} - R^a R^b\right) - iy^{*\mu}\sum_a \hat{y}_a R^a\right)$$

(102)

Now, in the limit $d \to \infty$, Eq. 101 can be approximated using the saddle-point approach (Bender & Orszag, 2013),

$$\langle\langle Z^n\rangle\rangle_{x,W} \approx \textbf{extr}_{Q_0,\hat{Q}_{0a},Q^{ab},\hat{Q}^{ab},R^a,\hat{R}^a} \exp\left(P\left[\frac{i}{2}\sum_a Q_0\hat{Q}_{0a} + i\sum_{a<b}Q^{ab}\hat{Q}^{ab}\right.\right.$$

$$\left.\left. + i\sum_a R^a\hat{R}^a + G_S(\hat{Q}_{0a}, \hat{Q}^{ab}, \hat{R}^a) + \alpha G_E(Q^{ab}, R^a)\right]\right)$$

(103)

where, **extr** corresponds to extremization of $\langle\langle Z^n\rangle\rangle_{x,W}$ over the respective order parameters. Performing this extremization over $\hat{Q}_{0a}, \hat{Q}^{ab}$ and $\hat{R}^a$, then generates an expression of the form,

$$\langle\langle Z^n\rangle\rangle_{x,W} = \textbf{extr}_{Q_0,Q,R}\exp\left\{nN\left(\frac{1}{2}\frac{Q-R^2}{Q_0-Q} + \frac{1}{2}\ln(Q_0-Q) - \frac{\alpha}{2}\ln\left[1+\beta(Q_0-Q)\right]\right.\right.$$

$$\left.\left. - \frac{\alpha\beta}{2}\frac{1-2R+Q}{1+\beta(Q_0-Q)}\right)\right\}$$

(104)

where we have invoked *replica symmetry* in the form, $Q^{ab} = Q$ and $R^a = R$, and that $\mathcal{E}_\mathcal{T} = (y^* - y)^2/2$. Plugging this back into Eq. **??**, then finally yields,

$$\beta f = -\textbf{extr}_{Q_0,Q,R}\left\{\frac{1}{2}\frac{Q-R^2}{Q_0-Q} + \frac{1}{2}\ln(Q_0-Q) - \frac{\alpha}{2}\ln\left[1+\beta(Q_0-Q)\right]\right.$$

$$\left. - \frac{\alpha\beta}{2}\frac{1-2R+Q}{1+\beta(Q_0-Q)}\right\}$$

(105)

The remaining pair of order parameters generate the following set of transcendental equations on extremization (Bös, 1998):

$$R = \frac{\alpha}{a}$$

$$Q = \frac{\alpha}{a^2 - \alpha}\left(1 - \frac{2-a}{a}\alpha\right)$$

$$Q_0 = Q + \frac{1}{\beta(a-1)}$$

(106)

where, $a = \max[1, \alpha]$ for $T \to 0$.

Now, the above determined values of $R, Q$ and $Q_0$ can be perceived as the *maximally likely* values of $R, Q$ and $Q_0$ of our teacher-student setup, for an inverse temperature $\beta$ parameterizing the system.

## C  EXTENDED EXPERIMENTS

Figure 4 presents the analytical generalization dynamics for two values of $\kappa$ and provides comparison between the theory and simulation results of the same model. We observe that the theory and

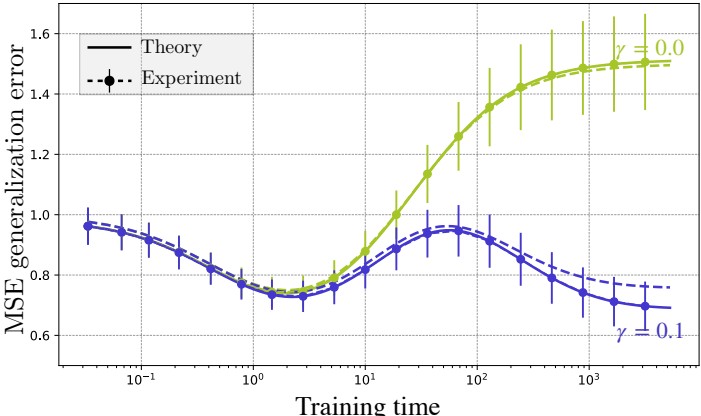

Figure 4: The teacher-student set-up in Sec. equation 2.1. We compare the analytical solutions to simulations performed on our teacher-student setup with $d = 100$, $p = 50$, $n = 150$ and we plot the error bars over 100 random seeds. The solutions and the simulations match closely and we observe double descent over the generalization error.

simulations accurately match. Further experiments are provided in the following anonymous Colab notebook.

Before diving into the theory, we invite the reader to recall a simple equation from thermodynamics. Consider an ideal gas in a container with its large number of molecules moving around, colliding with each other, all while obeying Newton's laws. While the exact dynamics of each of such molecules is intractable, the system's macroscopic behavior can be characterized in terms of a handful of scalar quantities, namely, the pressure $P$, the volume $V$, and the temperature $T$. By averaging over suitable probability measures and applying the principle of free-energy minimization, one arrives at a remarkably simple relationship between these three macroscopic variables, i.e., the well-known $PV = nRT$ ($n$: number of moles of gas, $R$: gas constant) (Reif, 2009).

