# OpenReview forum: "Multi-scale Feature Learning Dynamics: Insights for Double Descent"
_ICLR.cc/2022/Conference — ICLR 2022 Submitted_

### Official Review · Reviewer_tRFb · 2021-11-02

**Correctness:** 3
**Technical Novelty And Significance:** 2
**Empirical Novelty And Significance:** 2
**Recommendation:** 5
**Confidence:** 5

**Main Review:**

Strengths:

--The analysis of a simple model that exhibits epoch-wise double descent is illuminating and worthwhile, so the topic is of general interest.

--The concrete findings from the paper (exhibited in the numerics of Fig 2, Fig 3) help build intuition for the phenomenon; the agreement between finite-size experiments and analytic theory is good.




Weaknesses:

--Numerous typos in the main text and appendix that need to be addressed & divert the flow of the calculations for the reader. See more on this below. This also made it a bit challenging to check the calculations in full.

--I am not certain as to whether there are notable technical advancements in the paper. Relatedly, I am not positive if the intuition involved this paper (e.g. introducing multiple scales into the problem) has appeared before; for instance, can there be more discussion of how this paper relates to Heckel & Yilmiz (2020), as well as Stephenson & Lee (2021)?

--I think this paper could improve a bit on how the student-teacher model could be connected to the realistic setting (deep neural network with label noise on training data). Although I appreciate full analysis of a simple setting, I wonder if there is more that can be done (numerically or otherwise) to make a connection. Does the label noise effectively create two scales for the problem?

--I would like to better understand some of the prior results this work relies on for the analytic calculation; a self-contained discussion of the exact results drawn from older literature in the appendix would be helpful. For instance, regarding Eq. (17) and the probability distribution induced by SGD. Is this for a fixed choice of initialization, and considering the stochasticity / distributional nature arises from the Gaussian noise in SGD (Eq. 4)? In that case, how robust is this to other choices of noise? What justifies then the time-dependent distribution in Eq. 18, since there isn't a notion of equilibrium in the finite time case? Overall, I also think this paper could be clearer about how it treats the various sources of randomness (e.g. why it chooses to do averaging over SGD noise separately from the average over x and W).

--(Not a major source of weakness, but a suggestion.) Some of the writing and terminology seem too imprecise to be useful. For instance, I would suggest removing the first paragraph on Sec. 2.1 since it is somewhat tangential (discussion of microscopic & macroscopic quantities in the following paragraph are sufficient). The authors also write about the "interaction" of different feature learning speeds, which makes me think of a precise notion of interaction in the sense of physics, although I believe the authors simply mean the "presence" of different scales.

--The abstract mentions usage of tools from random matrix theory -- where does this appear?


Typos in manuscript & other comments:

--Eq.(5): Since there is an expectation over the teacher noise \epsilon, why isn't the loss in terms of (y-\hat{y}) instead of (y* - \hat{y})?

--It is common to average over the draw of finite-size training dataset. I am not sure how this appears in Eq. 5; could the authors comment on this? I understand E_{x} to be over the population distribution on x, and E_{W, \epsilon} average over the choice of teacher model.

--Above Eq. 8, it is written X = F^T Z, but X = ZF is used in Eq. 8.

--Eq. (20) there is a reference to W* but I do not find it introduced earlier. (It seems these are the teacher weights, W -> W*?)

--Figure 2: a/b/c/d figures are mislabeled relative to the captions. In caption (b), k -> \kappa.

--Eq. (23): from the first to second line, how did z -> x without an appearance of F?

--In Eq. (24)/(25), the average over x gives rise to Kronecker delta orthogonality; since z ~ Normal, how does this hold for generic F? Is there an assumption of orthogonality on F?

--Eq. (28): Z^n should be labeled Z^r.

--Eq. (29): what is x^*? As with earlier equations, I assume this equation refers to the teacher network but don't see input z referenced.

--Eq. (30): capital N is introduced. \mu index also previously referenced training sample index (up to n).

--Eq. (31): usage of capital P, is this same as lowercase p?

--Eq. (35): two lines are copied identically.




**Summary Of The Paper:**

This paper theoretically analyzes a student-teacher setting for which epoch-wise double descent occurs. (Epoch-wise double descent, previously discussed in the literature, refers to the non-monotonicity of the population error as a function of training time.) More specifically, the setup is: (a) a linear teacher with Gaussian noise; (b) a linear student whose inputs are a linear transformation F ("modulation matrix") applied to the teacher's inputs. The inputs are d-dimensional and there are n training samples. Exact analytic expressions for the test error can be derived by utilizing the replica method in the limit of n, d -> \infty. The case analyzed in most depth is a setting in which F has two scales for its singular values (with high degeneracy) -- c.f. Assumption 1. Both the theory and finite-size simulations indeed exhibit epoch-wise double descent depending on the choice of setting parameters (e.g. strength of regularization and condition number of F). The intuition behind this is that when the scales are well-separated and learnable (e.g. regularization not too strong), overfitting of the faster features occurs (leading to a rise in the test error) before learning of the slower features (which subsequently decreases the test error).

**Summary Of The Review:**

My slightly lower score is based on the following factors:
--Since the focus of this paper is not empirical (e.g. the observation in realistic networks has appeared before), the contributions come primarily from the theoretical side. In this respect, I am not sure if similar insights in linear models & control of multiple scales for epoch-wise double descent have appeared in earlier works and would appreciate if the authors could comment on this in detail.
--The typos / errors made the calculations more ambiguous and harder to assess the correctness of the final result.

I would be happy to consider adjusting my score based on the author's response.

---

> ### Author Response · Authors · 2021-11-17
> **Author Response [Part 1]**
>
> We are happy that you have found our submission illuminating and thank you for the constructive comments. Below, we do our best to address your questions.
>
> > Numerous typos that divert the flow of the calculations for the reader.
>
> We apologize for the typos and errors that have slipped through. We have made sure that the errors are resolved. You may use the “compare” tool of the openreview to view the changes.
>
> > I am not certain as to whether there are notable technical advancements in the paper. Relatedly, I am not positive if the intuition involved this paper (e.g. introducing multiple scales into the problem) has appeared before; for instance, can there be more discussion of how this paper relates to Heckel & Yilmaz (2020), as well as Stephenson & Lee (2021)?
>
> We thank the Reviewer for their question and would like to apologize if the technical advancements of our work are not overtly clear in comparison to the related works mentioned. Below, we list a number of clarification points that are also incorporated in the revised paper. We are confident that the following adequately states our contribution in comparison to the current literature, as well as establish key novel points we contribute.
>
> First, we provide some context about our statistical physics approach. The statistical physics-based replica method is a powerful tool that has been used in the context of neural networks to study their “asymptotic” ($t \rightarrow \infty$) generalization performance. Specifically, the model-wise double descent was predicted in the 90's (See for example Fig. 5.9 of [[Opper and Kinzel 1996]](https://journals.aps.org/pre/pdf/10.1103/PhysRevE.47.1384?casa_token=Gd7AZkVrlLkAAAAA%3ApJ_9omXWaIXSz0FUIFADyoFUL_xpVQC1TsByeiQLDwTf38ndzeu3As3sZyGO16w6w-wYrEWVbO_XXso3) on page 36).
>
> Despite its effectiveness, this framework has the key limitation that it requires various system parameters (including time) to go to infinity (the asymptotic condition). This limitation has prevented researchers from studying the “finite-time dynamics”, and thus, transient phenomena such as epoch-wise DD. As such, our major novel contribution is to adapt the statistical physics approach developed to study model-wise learning phenomena, to the epoch-wise setting. This is performed through the usage of links between early-stopping and ridge regularization (see our response to Reviewer ZC59) along with the incorporation of anisotropic input features.
>
> This builds and strengthens recent work that explored this phenomenon.  Indeed, as discussed on Page 9, our findings and those of [[Heckel & Yilmaz (2020)]](https://arxiv.org/abs/2007.10099) and [[Stephenson & Lee (2021)]](https://arxiv.org/abs/2108.12006) reinforce one another with a common central finding that the epoch-wise double descent results from different features being learned at different time-scale. In these two related papers, the authors build on random matrix theory to study the generalization performance, with the latter providing only an integral equation for the test error rather than a closed-form expression. Our approach, using the replica technique on the other hand allows us to describe the “high-dimensional emergent properties” through a set of “low-dimensional scalar variables”, which can be used for studying other generalization behaviors of neural networks.
>
> Importantly, these related works also use different data models; specifically, in [[Stephenson & Lee (2021)]](https://arxiv.org/abs/2108.12006), the data model is constructed so that the noise is explicitly added *only* to the fast-learning features while slow-learning features remain noise-free. Such a data model also exhibits epoch-wise double descent, though it might appear less justified than feature differences that arise from heterogeneity in the input statistical structure, a setting our model captures. Nevertheless, we believe that our work contributes significantly to elucidating epoch-wise DD, precisely because it arrives at conclusions consistent with prior results, in a setting with increased data structure modelling and a focus on learning dynamics.
>
> Finally, we believe our theoretical framework sets the stage for further understanding of generalization dynamics *beyond* the double descent. A future direction is to study a case in which the first descent is strong enough to bring down the training loss to very small values to the point that learning slower features is practically impossible or happens after a very large number of epochs. That could result in models being biased towards learning more superficial features and failing to discover more complicated generalizable features.
>
> These clarification points are included in the revised manuscript at Section 4.

---

> > ### Author Response · Authors · 2021-11-17
> > **Author Response [Part 2]**
> >
> > > I think this paper could improve a bit on how the student-teacher model could be connected to the realistic setting (deep neural network with label noise on training data). Although I appreciate full analysis of a simple setting, I wonder if there is more that can be done (numerically or otherwise) to make a connection. Does the label noise effectively create two scales for the problem?
> >
> > We share the Reviewer’s point of view here. Our student-teacher model is built upon the common intuition in neural networks that “not all features are learned at the same speed”. For example, it is well-known that convolutional networks are biased towards learning color and texture much faster than other features [[Geirhos et al. (2018)]](https://arxiv.org/abs/1811.12231). That is while the underlying data generating process, i.e., nature/teacher, might put more emphasis on harder-to-learn features. This mismatch between the importance of features and their speed of learning is in fact, modeled through the modulation matrix $F$ such that smaller singular values of $F$ correspond to slower features and vice versa. We believe that this central attribute is captured by our simple model, but we also acknowledge that its linear nature (which enables tractable analysis) differs from that of widely used neural networks. For a non-linear neural network, the notion of the matrix $F$ is not readily present, instead relying on learned implicit transformations.
> >
> > However, linearizing a neural network using the Neural Tangent Kernel (NTK) approximation [[Jacot et al. (2018)]](https://arxiv.org/abs/1806.07572) enables us to draw insightful parallels in certain regimes. We have,
> > $\hat{y} := f_{\theta}(X)$
> > $\hat{y} \approx \Phi \theta \quad \text{where,} \quad \Phi := (\nabla_{\theta} \hat{y})^T$
> > The matrix $\Phi^T \Phi$ is called the NTK Gram matrix and echoes $F^T F$ in our teacher-student setup. Consequently, to answer the reviewer's question on whether we also observe two scales in neural networks, we can look at the singular values of $\Phi^T \Phi$.
> >
> > Crucially, building on the reviewer’s question and to verify if deep networks exhibit learning dynamics that are consistent with the approximation described above, we study a ResNet18 trained on Cifar-10 which exhibits double descent. We apply an NTK approximation and plot the singular values of the NTK Gram matrix as well as the norm of their associated weight vector. [This Figure](https://raw.githubusercontent.com/NNdoubledescent/doubledescent/main/extra_experiments/rebuttal.png) summarizes the results:
> > SubFigure (a): A bipartitioning is observed in singular values of $\Phi^T \Phi$ such that the first ~1000 singular values are relatively large while the rest are significantly smaller.
> > SubFigure (b): Immediately after the first descent, we observe that the weights connecting to larger singular values are learned.
> > SubFigure (c): After the second descent, we observe that the remaining weights connecting to smaller singular values are learned as well.
> > Analysis: We conjecture that the first descent in the generalization error can be attributed to learning of fast features (those with large singular values) while the subsequent descent can be attributed to learning of slow features.
> >
> > Finally, we would like to reiterate that the current submission falls into a category of work at the intersection of theory and practice. A central idea here is to introduce simple models that exhibit interesting properties of neural networks yet allow for the analytical study and provide insights. To this end, our proposed teacher-student model is a step towards such a goal. However, further analysis/experiments are still required to validate the transferability of findings in simple linear models to neural networks.
> >
> > > A self-contained discussion of the exact results drawn from older literature in the appendix would be helpful.
> >
> > We have updated the appendix accordingly to provide clear references to the literature we build upon.
> > Particularly, in the revised version, in App. B.1, we have provided a self-contained proof of Eq. 6 (the decomposition of the generalization loss).
> > In App. B.2, we provide a step-by-step proof of the results for the general case.
> > In App. B.3, we have added more fine-grained derivations of the free-energy and subsequent macroscopic equations for two components of the student model.

---

> > > ### Author Response · Authors · 2021-11-17
> > > **Author Response [Part 3]**
> > >
> > > > Regarding Eq. (17) and the probability distribution induced by SGD. Is this for a fixed choice of initialization, and considering the stochasticity / distributional nature arises from the Gaussian noise in SGD (Eq. 4)? How robust is this to other choices of noise?
> > >
> > > You have raised an important question that has been the subject of research in recent years [see for example the [“Non-Gaussianity of Stochastic Gradient Noise” by Panigrahi et al. (2019) and references there](https://arxiv.org/abs/1910.09626)]
> > >
> > > However, it is commonplace to approximate SGD as a stochastic differential equation and it relies on the assumption that the noise of SGD is isotropic Gaussian [[Bottou (1991)](https://leon.bottou.org/publications/pdf/nimes-1991.pdf), [Mandt et al. (2017)](https://arxiv.org/abs/1704.04289), [Li et al. (2017)](https://proceedings.mlr.press/v70/li17f.html)]. While extension to other choices of noise is not trivial and outside the scope of our work, [[Jingfeng et al. (2019)](https://arxiv.org/pdf/1906.07405.pdf)] suggest that thanks to the central limit theorem, the assumption of Gaussian noise is reasonable.
> > >
> > > > What justifies then the time-dependent distribution in Eq. 18, since there isn't a notion of equilibrium in the finite time case?
> > >
> > > We require a time-dependent distribution to study the transient dynamics. Previous related work [[Kuhn & Bos (1993)](https://iopscience.iop.org/article/10.1088/0305-4470/26/4/012), [Gerace et al. (2020](https://arxiv.org/abs/2002.09339)] only describe the equilibrium distribution, and as you correctly pointed out, there is no notion of equilibrium in finite time. Therefore, we introduce a modified alternative loss $\tilde{\mathcal{L}}$ in Eqs.  20-23 such that its equilibrium is the same as the solution of SGD after $t$ iterations on the original loss $\mathcal{L}$. The newly added schematic plot on page 6 depicts the relationship between $\mathcal{L}$ and $\tilde{\mathcal{L}}$.
> > >
> > > > Overall, I also think this paper could be clearer about how it treats the various sources of randomness (e.g. why it chooses to do averaging over SGD noise separately from the average over x and W).
> > >
> > > We acknowledge that our discussion of the various noise sources in our setup can be improved and intend to ensure that. We would though like to clarify that we perform averaging over the input distribution _x_, the teacher weight configuration _W_, and input noise only. The stochasticity of SGD on the other hand can be understood to be regulating the temperature parameter $\beta^{-1} \equiv T$ in our analytical framework thus governing the student weight distribution at the end of training. For example, if the SGD noise is assumed to be Gaussian ($N(0, \sigma^2)$) for the discussion, then $\sigma^2 = \frac{1}{\beta^2}$.
> > >
> > > > Some of the writing and terminology seem too imprecise to be useful. For instance, I would suggest removing the first paragraph on Sec. 2.1 since it is somewhat tangential (discussion of microscopic & macroscopic quantities in the following paragraph are sufficient). The authors also write about the "interaction" of different feature learning speeds, which makes me think of a precise notion of interaction in the sense of physics, although I believe the authors simply mean the "presence" of different scales. - The abstract mentions usage of tools from random matrix theory -- where does this appear?
> > >
> > > We apologize for instances of inaccurate use of words. We have updated the paper accordingly so that ambiguous words are replaced, the opening of  Sec. 2.1 is moved to the appendix, and flow of presenting the analytical results is improved.
> > >
> > > Q6. Other comments.
> > > > Eq. 5: [...] an expectation over the teacher noise \epsilon [...].
> > >
> > > You are right! The expectation is now updated to be only wrt to the teacher’s input distribution z.
> > >
> > > > It is common to average over the draw of a finite-size training dataset. I am not sure how this appears in Eq. 5; could the authors comment on this?
> > >
> > > Eq. 5 computes the “generalization error” and therefore it is an exact expectation. The training loss, however, is computed using a finite-size training dataset, hence the empirical expectation in Eq. 3 is over n datapoints.
> > >
> > > > Above Eq. 8, it is written X = F^T Z, but X = ZF is used in Eq. 8.
> > >
> > > You are right! The error is now fixed and we consistently use x := F^T z.
> > >
> > > > Eq. (20) there is a reference to W* but I do not find it introduced earlier.
> > >
> > > You are right! It is now updated.
> > >
> > > > Figure 2: a/b/c/d figures are mislabeled relative to the captions. In caption (b), k -> \kappa.
> > >
> > > You are right! The figure is updated.
> > >
> > > > Eq. (23): from the first to second line, how did z -> x without an appearance of F?
> > >
> > > You are right! There has been a mix-up between x and z. The derivation is now fixed, but we  note that the final expression for this equation remains unchanged.

---

> > > > ### Author Response · Authors · 2021-11-17
> > > > **Author Response [Part 4]**
> > > >
> > > > > In Eq. (24)/(25), the average over x gives rise to Kronecker delta orthogonality; since z ~ Normal, how does this hold for generic F? Is there an assumption of orthogonality on F?
> > > >
> > > > Once again, the error here is due to the mix-up of x and z. The expectation is wrt z and z is assumed to be Gaussian.
> > > >
> > > > > Eq. (28): Z^n should be labeled Z^r. / Eq. (29): what is x^*? / Capital N and P are introduced. / Eq. (35): two lines are copied identically.
> > > >
> > > > Thank you very much for your thorough review. We have updated the whole appendix section. We have made sure that the notation is consistent, matrices commute properly, and the flow is smooth.
> > > >
> > > > Once again, we appreciate your time. Please let us know if any further clarification is needed.

---

### Official Review · Reviewer_ZC59 · 2021-11-02

**Correctness:** 3
**Technical Novelty And Significance:** 3
**Empirical Novelty And Significance:** 3
**Recommendation:** 8
**Confidence:** 4

**Main Review:**

My major concerns include:
1. The implicit regularization effects of SGD/GD on least square linear regression may perform as a Ridge regularization. (Ali and Ray Tibshrani's work at ICML 2020), where the number of learning epochs/steps are connected to the inverse of lambda (strength of Ridge effects). It is not superise that when you tune the lambda of Ridge, certain double descent would appear in testing accuracy.  Shall authors discuss the connections between this work and Rayn Tibshirani's works on ICML 2020 and AISTATS 2019 on Ridge-style implicit regularization of GD and SGD for OLS?

2. whether authors tried to connec their work to the linear regression problem under HDLSS settings (d>>p). In such setting, the analytical results of SGD convergence might be different (inclusion of pseudo inverse). In that sense, the result would lead to Ridgeless regression. There are quite a lot of works studying the double descent of Ridgeless. Shall authors discuss the connections between this work and those works?

**Summary Of The Paper:**

This work studied epoch-wise double descent using linear model as a proxy. Basically, authors proposed a linear teacher-student models to established analysis and used random matrix theory (rmt) to interpret the learning dynamics. Some simulation on the well-posed (n>p) linear regression problems backups the theory. Some real-world experiments based on ResNet-18 and noisy labels also demonstrate the relevance of proposed theorem.

**Summary Of The Review:**

It is a solid work with theoretical analysis and empirical evaluation. Though it may connect to many pioneering works in the field, authors should discuss these connections.

---

> ### Author Response · Authors · 2021-11-17
> **Author Response**
>
> We are delighted to hear that you have found our submission solid. Below, we do our best to address your questions.
>
> > Shall the authors discuss the connections between this work and Ali et al. (2019) and Ali et al. (2020) on the ridge-style implicit regularization of GD and SGD for OLS?
>
> Thanks for the above question. The results presented in our work have a core dependence on the results presented in Ali et al. (2019), (2020) and we clarify this point in the revised version. To the best of our knowledge, these works first formalized the connection between (continuous-time) GD or SGD-based training of an OLS setup and that of ridge regression, providing bounds on the test error under these algorithms over training time $t$, in terms of a ridge setup with ridge parameter $\lambda = 1/t$. We utilize these results in the sense that, by evaluating the generalization error $\mathcal{L}_G$ of our student-teacher setup with explicit ridge regularization, we invoke the connection between the ridge coefficient $\lambda$ and training time $t$ as described in these works, to obtain the behavior of (ridgeless) $\mathcal{L}_G$ over training. This determination of an expression of $\mathcal{L}_G (t)$ is what allows us to study the epoch-wise DD phenomenon.
>
> In fact, to study the epoch-wise generalization properties of our model, we require a time-dependent distribution. Previous related work [[Kuhn & Bos (1993)](https://iopscience.iop.org/article/10.1088/0305-4470/26/4/012), [Gerace et al. (2020](https://arxiv.org/abs/2002.09339)] only describe the equilibrium distribution($t \rightarrow \infty$). That is while, there is no notion of equilibrium in finite time. Therefore, we introduce a modified alternative loss $\tilde{\mathcal{L}}$ in Eqs. 20-23 such that its equilibrium is the same as the solution of SGD after $t$ iterations on the original loss $\mathcal{L}$. The newly added schematic plot on page 6 depicts the relationship between $\mathcal{L}$ and $\tilde{\mathcal{L}}$. We have included a discussion on this matter in the App. A of the revised version. To validate the results empirically, we provide an [anonymous colab notebook](https://colab.research.google.com/drive/19OihPp2jfs6TpyEGg92eXjo3d4NjNW2I?usp=sharing#scrollTo=vyHLls8lRq1G) which compares the empirical results using $\mathcal{L}$ and $\tilde{\mathcal{L}}$.
>
> > Whether the authors tried to connect their work to the linear regression problem under HDLSS settings (d>>p). In such a setting, the analytical results of SGD convergence might be different (inclusion of pseudo inverse). In that sense, the result would lead to Ridgeless regression. There are quite a lot of works studying the double descent of Ridgeless. Shall authors discuss the connections between this work and those works?
>
> Thanks for raising this point. As essentially explained in the response to the previous question, our derivation introduces an implicit correspondence whereby the eventual substitution of $\lambda = 1/(\eta t)$ translates the test error result to the ridgeless problem evaluated at training time _t_. We found that it is not straightforward to dissociate ridge regularization from training dynamics in our setup, and would leave it to future work to elucidate on this.
>
> > It is a solid work with theoretical analysis and empirical evaluation. Though it may connect to many pioneering works in the field, authors should discuss these connections.
>
> Once again, thank you for your comments. We have added a section in Appendix A at which we have extended our discussions on additional related work.
>
> We appreciate your time. Please let us know if any further clarification is needed.

---

### Official Review · Reviewer_Tczc · 2021-11-03

**Correctness:** 4
**Technical Novelty And Significance:** 3
**Empirical Novelty And Significance:** 3
**Recommendation:** 8
**Confidence:** 4

**Main Review:**

This insightful work demonstrates that another apparently exotic behavior of deep neural networks, non-monotonic generalization error over the course of training, is already present in simple, analytically tractable linear models. The paper is clearly written, the theory is well-motivated, and the results are neatly presented. I only wish that the authors dove more deeply into interpreting the behavior of their analytical theory: (1) what happens when features of more than two scales are present? is there triple descent in the generalization error? (2) how does epoch-wise double descent differ from model-size double descent? one difference seems to me that linear teacher-student models trained on isotropic data exhibit model-size double descent but apparently not epoch-wise double descent, which requires anisotropic data. What explains the difference? Moreover, the peak in the model-size double descent curve has a nice interpretation as the model size necessary to perfectly interpolate the training data. Is there a similar interpretation for the peak of the epoch-wise descent curve? More broadly, epoch-wise double descent is sometimes explained by training time controlling model complexity in analogy to model-size double descent - but this work appears to suggest a different mechanism. It would be interesting to discuss this further. (3) How do the results change as a function of the the overlap between the teacher and the anisotropy in the data?

Note: I believe the color code in Fig. 2 is incorrect, and is misleading. I think the colors of the large and intermediate regularization strength curves should be flipped.

**Summary Of The Paper:**

An influential line of work has revealed that deep neural networks can exhibit non-monotonic behavior in their generalization error (double descent) as a function of model size, dataset size, and training time. Recent (and old) theoretical work has demonstrated that even simple models like linear regression exhibit the same non-monotonic behavior as a function of model and dataset size. The authors of this work build on this literature by demonstrating that linear teacher-student models trained with gradient descent can exhibit double descent as  a function of training time. Using replica theory, they derive a closed-form expression for the generalization error of this model as a function of training time. They show that double descent can arise when the student is trained on anisotropic data which includes a set of high SNR features and a set of low SNR features. Finally, the authors demonstrate a qualitative match between the generalization error behavior of linear teacher-student regression and a ResNet18 trained on CIFAR10 as a function of training time and regularization strength.

**Summary Of The Review:**

I think this work is a valuable contribution which captures an interesting feature of the training dynamics of generalization error in deep neural networks in a simple, analytically tractable model. The close qualitative match between the behavior of the simple model and a ResNet on CIFAR10 suggests that the mechanisms identified here may be general, and the theory derived by the authors lays the groundwork for deeper investigations into the training dynamics of generalization error in neural networks.

---

> ### Author Response · Authors · 2021-11-17
> **Author Response**
>
> We are thrilled to hear that you have found our submission insightful and well-motivated. Below, we attempt to address your questions.
>
> > (1) what happens when features of more than two scales are present? Is there triple descent in the generalization error?
>
> This is an interesting question. While incorporating multiple scales in our analytical framework was a challenge, we do nevertheless believe that the presence of the same does lead to multiple descents. Inspired by your question, we conducted an experiment in which the modulation matrix $F$ has three distinct scales. As illustrated in [this anonymous Google Colab notebook](https://colab.research.google.com/drive/1N9JuUzsnkYKemiHpVEvXK5kj5jvcEb-m?usp=sharing), we are able to design a dataset for which a triple-descent generalization curve is observed. Although we would like to draw a distinction between such behavior and epoch-wise DD as observed in a DNN setting [Nakkiran et al. (2019)](https://arxiv.org/pdf/1912.02292.pdf), wherein the latter as in its nomenclature, one observes two descents only. It is an open problem and scope for future research to determine the cause(s) behind such an observation.
>
> > How does epoch-wise double descent differ from model-wise double descent? One difference seems to me that linear teacher-student models trained on isotropic data exhibit model-size double descent but apparently not epoch-wise double descent, which requires anisotropic data. What explains the difference?
>
> In accordance with its name, model-wise double descent (in the test error) occurs due to an increase in model size (number of its parameters) i.e. as the model transitions from an under-parameterized to an over-parameterized regime. A variety of works have tried to understand this phenomenon from the lens of implicit regularization [Neyshabur et al. (2014)](https://arxiv.org/pdf/1412.6614.pdf), defining novel complexity measures [Neyshabur et al. (2017)](https://arxiv.org/pdf/1706.08947.pdf) etc. On the other hand, epoch-wise double descent (in the test error) as treated in our work, is observed to occur for both over-parameterized [Nakkiran et al. (2019)](https://arxiv.org/pdf/1912.02292.pdf) and under-parameterized [Heckel and Yilmaz (2020)](https://arxiv.org/pdf/2007.10099.pdf) setups. As found in our work along with the latter reference, this phenomenon seems to be a result of different feature learning speeds rather than the extent of model parameterization. The overlap of the test-error contributions from the different weights with varying scales of learning henceforth leads to a non-monotonous evolution of the model test error as exemplified by epoch-wise double descent. We clarify this distinction in App. A in the revised text.
>
> > The peak in the model-size double descent curve has a nice interpretation as the model size is necessary to perfectly interpolate the training data. Is there a similar interpretation for the peak of the epoch-wise descent curve?
>
> Thank you for this insightful question. While the model-wise DD peak is associated with the model’s capacity to perfectly interpolate the data, we do not think an analogous notion exists for the case of epoch-wise DD. Our understanding of the peak in the latter is that it corresponds to a training time configuration whereby a subclass of features are already learned (due to a larger associated signal-to-noise ratio) and are being overfitted upon to fit the target. As training proceeds further, the remaining set of features are eventually learned thus allowing for a lowering of the test error. We have included this discussion in the App. A of the revised text.
>
> > More broadly, epoch-wise double descent is sometimes explained by training time controlling model complexity in analogy to model-size double descent - but this work appears to suggest a different mechanism. It would be interesting to discuss this further.
>
> Thanks again for raising this important point. To the best of our understanding, the seeming connection between model-wise and epoch-wise double descent, by viewing the latter as a mechanism arising due to a training time-controlled model-complexity (EMC: effective model complexity), might not be entirely valid [see Sec. 8 of [Stephenson &  Lee (2021)](https://arxiv.org/pdf/2108.12006.pdf) on EMC hypothesis]. The distinction between the two can be understood as a result of model overfitting (model-wise) vs. overtraining (epoch-wise). While increasing the parameters and therefore the complexity, causes model-wise DD, epoch-wise DD occurs while holding the model size and hence the ‘total complexity’ fixed, and training exhaustively such that all the features are fully learned (within a setup of different feature learning scales).

---

> > ### Author Response · Authors · 2021-11-17
> > **Author Response [Continued]**
> >
> > > How do the results change as a function of the overlap between the teacher and the anisotropy in the data?
> >
> > We believe that the question is how are our results affected if the teacher network itself also accesses the data via an analogous modulation matrix $F$ and hence observes anisotropic data. In such a case, while our results (e.g. $R$ and $Q$ expressions) will qualitatively change, the epoch-wise DD phenomenon will not be observed **unless** the student network itself accepts anisotropic data, thus enforcing different feature learning scales. In other words, having a teacher-student model of the form: $y^* = z^T F’ W$ and $\hat{y} = z^T \hat{W}$ (isotropic data access by the student), can be perceived as the student learning a target function defined by an effective weight configuration $F’ W$ albeit at the same rate.
> >
> > > I believe the color code in Fig. 2 is incorrect and is misleading. I think the colors of the large and intermediate regularization strength curves should be flipped.
> >
> > We apologize if the color-coding was misleading. We have updated the figure accordingly in the revised version.
> >
> > Once again, we appreciate your time. Please let us know if any further clarification is needed.

---

### Public Comment · ~Denny_Wu1 · 2021-11-17
**Question**

Thanks for the interesting work. I have one clarifying question:
For the substitution from Equation (18) to Equation (19), why is the $L_2$-regularized risk (which we know how to compute from prior works) asymptotically the same as the gradient descent risk? What is the meaning of "$\approx$" in Equation (19)?

I'm curious because in Section 4.2 of (Amari et al. 2020), we tried to compute the asymptotic risk of gradient flow in a similar setting. However, it was clear to us that the precise risk does not have a convenient closed-form, so we were only able to prove monotonicity of the bias and variance with respect to training time in special cases (see Appendix D).

Amari et al. 2020. https://openreview.net/pdf?id=S724o4_WB3.

---

> ### Author Response · Authors · 2021-11-29
> **Connection between L2-regularized risk and gradient descent risk.**
>
> Dear Denny,
>
> Thank you for your interest and sorry for the late reply!\
> Gradient descent dynamics admit an exact solution as in Eqs. 9, 10, however, as you correctly pointed out, it is not convenient as it requires computation of the eigenvalues of $X^TX$.\
> As a result, we also study the special case in Section 2.2 and use an approximation based on the taylor series of $ln(1+x) \approx x$.
> We introduce a modified alternative loss $\tilde{\mathcal{L}}$ in Eqs. 20-23 (the current revised version) such that its equilibrium is the same as the solution of SGD after $t$ iterations on the original loss $\mathcal{L}$. The newly added schematic plot on page 6 depicts the relationship between $\mathcal{L}$ and $\tilde{\mathcal{L}}$. We have included a discussion on this matter in the App. A of the revised version.
>
> You might want also to take a look at [this colab notebook](https://colab.research.google.com/drive/19OihPp2jfs6TpyEGg92eXjo3d4NjNW2I?usp=sharing#scrollTo=vyHLls8lRq1G) which compares the empirical results using $\mathcal{L}$ and $\tilde{\mathcal{L}}$.
>
> We have also added the a proof to the appendix which we will include in the camera ready version: [link](https://github.com/NNdoubledescent/doubledescent/blob/main/extra_experiments/proof_page_21.pdf).
>
> Hope that answers your question!

---

### Author Response · Authors · 2021-11-17
**Thank you for your comments.**

We would like to thank all the reviewers for their valuable feedback and for the kind support of this work.  The reviews contributed to genuinely improve this paper, especially by pointing out the need for clarifications concerning our theoretical analysis, its relation to other existing work in the literature, its applicability to real world applications, and by pointing out some typographical errors. We are confident that we address all major concerns from reviewers in the revised manuscript, and outline the precise changes made in detailed response to reviewer comments.

---

### Author Response · Authors · 2021-11-29
**End of the discussion period.**

Dear everyone,

We would like to thank the reviewers and the AC for their time. We found all reviews constructive and fair. Although the reviewers did not engage in the discussion period, we are confident that we have addressed all major concerns. Here we provide a concise list of the raised concerns and the actions we have taken to address them:

> Reviewer tRFb: What are the technical advancements? How this relates to Heckel & Yilmaz (2020) and Stephenson & Lee (2021)?

**Action** :
- On Section 2.2, "Induced probability density of SGD" is updated, and a new figure is added to pinpoint the difficulty of derivations and their novelties.
- Paragraphs 4, 5, 6 on page 9 are updated/added to relate our findings with those of Heckel & Yilmaz (2020) and Stephenson & Lee (2021).

> Reviewer tRFb: Connection to realistic settings/deep networks?

**Action** : Further connection with NTK is provided along with a [new experiment](https://raw.githubusercontent.com/NNdoubledescent/doubledescent/main/extra_experiments/rebuttal.png) showcasing the emergence of two scales in deep networks.

> Reviewer tRFb: What justifies then the time-dependent distribution in Eq. 18?

**Action** : Clarifications are provided to the reviewer, and Section "Induced probability density of SGD" is updated to resolve any confusion.

> Reviewer tRFb: Some of the writing is imprecise.

**Action** : With several passes, we have made sure to use consistent and precise terminologies in the revised version.

> Reviewer tRFb: A self-contained discussion of the exact results drawn from older literature in the appendix would be helpful.

**Action** : Five new pages are added in Appendices B1 to B5, providing detailed standalone derivations of results used from older literature.

> Reviewer ZC59: Connection between implicit regularization of SGD and ridge-regularized loss?

**Action** : On pages 14-15, a discussion is added with adequate references. The equations 22-23 are updated, and on page 6, a new figure is added to showcase this connection. [Further comparisons](https://colab.research.google.com/drive/19OihPp2jfs6TpyEGg92eXjo3d4NjNW2I?usp=sharing#scrollTo=vyHLls8lRq1G) are also provided.

> Reviewer ZC59: Connection to pioneering work.

**Action** : A new discussion section is added in Appendix A, covering further related work.

> Reviewer Tczc: Is there triple descent in the generalization error?

**Action** : [Further experiments](https://colab.research.google.com/drive/1N9JuUzsnkYKemiHpVEvXK5kj5jvcEb-m?usp=sharing) are provided that show triple descent.

> Reviewer Tczc: the difference between epoch-wise and model-wise?

**Action** : On page 14, a discussion is added on that matter.

Minor changes:
- Typos are fixed, and the writing is improved.
- Misleading color codings are replaced.
- Notations are made consistent throughout the text.

We appreciate your consideration!

---

### Decision · Program_Chairs · 2022-01-20

**Decision:**

Reject

**Comment:**

This paper examines the time-dependent generalization behavior of high-dimensional student-teacher linear regression models. It introduces a simple two-scale covariance model and examines the exact solutions for the dynamics, finding a tradeoff between the fast- and slow-learning features, leading to epoch-wise double descent. Qualitative comparisons are made with the SGD dynamics of ResNe18 on CIFAR-10.

The reviewers offer split opinions on this work, with most reviewers finding strength in exhibiting the complex behavior of epoch-wise double descent in a simple and analytically-tractable setting. Weaknesses highlighted in the discussion include clarity, discussion of prior work, and rigor of the analyses.

I believe a clear demonstration and analytical explanation for epoch-wise double-descent would certainly be of interest to the ICLR community, and I concur with the reviewers who emphasize these strengths of the paper. However, as one reviewer mentioned, this paper is primarily a theoretical work, and as such, the main theoretical advancements over prior work should be clear, and the novel results should be sufficiently rigorous. In this regard, the paper is lacking, as detailed below.

First of all, the discussion of SGD is imprecise, with no explicit definition of the optimization method that is actually being performed. What is the batch size? How is the sampling performed? What is the learning rate/schedule? The formulas in Secs. 2.1-2.2 suggest that full-batch gradient descent is being performed. In Sec. 2.3, stochasticity from SGD is induced via a Gibbs distribution. However, contrary to the discussion, I don't think that this is a "well-known" **result** (though of course it is a well-known **model**), and in high-dimensions I am not sure it is even correct (see e.g. [1]).

Second of all, even assuming the Gibbs distribution, the substitution on line (23) is only justified in words, whereas the cited results from Ali et al., 2020, only provide a bound. What is meant by "$\approx$"? Some discussion is given about this step of the derivation, but more precise statements would really help make the argument convincing.

Finally, the derivations seem to rely on the replica method from statistical physics, which is not rigorous. While I am generally supportive of such methods for technically challenging problems that do not readily admit alternative analyses, given the simplicity of the linear model setup here, I believe a more rigorous approach would not be prohibitively difficult. At the very least, some acknowledgement should be given about the lack of rigor in the derivation.

Overall, this paper presents a simple and analytically tractable model that sheds light on the importance phenomenon of epoch-wise double descent. Unfortunately, the presentation is not sufficiently clear and the derivations not sufficiently rigorous to merit publication at this time.

[1] Paquette, Courtney et al. “SGD in the Large: Average-case Analysis, Asymptotics, and Stepsize Criticality.” COLT (2021).